# Beyond Reactivity: Proactive Adaptive Conformal Inference for Online LLM Factuality

Xinyu Liu [1]   Jun Wu [1]

## Abstract

Large Language Models (LLMs) often produce hallucinated outputs, which limit their reliability in high-stakes applications. Conformal prediction can provide guarantees on the correctness and factuality of LLM outputs, but existing approaches rely on the exchangeability assumption, which rarely holds in online settings where user queries and interests change over time. To solve this problem, in this paper, we propose **PACE** (**P**roactive **A**daptive **C**onformal Inferenc**E**), a novel framework that sequentially updates the time-varying target miscoverage parameter with a dynamic step size to maintain valid coverage under online distribution shifts. PACE is motivated by the theoretical connections between expected miscoverage error and key factors such as distribution shifts and instantaneous parameter error. It integrates two complementary signals: (1) a proactive shift detection to estimate the magnitude of distribution shifts, and (2) a reactive error that scales updates according to the local coverage gap. Extensive experiments on synthetic and real-world datasets demonstrate that PACE consistently outperforms advanced adaptive baselines. It reduces the deviation from the target error rate by up to 60% in QA tasks and accelerates coverage recovery by over 2.5x during abrupt shifts, ensuring stable factuality guarantees without compromising utility and stability.

**Conflict of Interest Disclosure.**   This work was supported in part by an Amazon Research Award. The authors declare no other competing financial interests related to this work.

---

[1] Department of Computer Science and Engineering, Michigan State University, East Lansing, Michigan, USA. Correspondence to: Jun Wu <wujun4@msu.edu>.

*Proceedings of the 43$^{rd}$ International Conference on Machine Learning*, Seoul, South Korea. PMLR 306, 2026. Copyright 2026 by the author(s).

## 1. Introduction

Large Language Models (LLMs) have demonstrated remarkable capabilities across a variety of tasks, yet their deployment in high-stakes domains involves significant risk (Wiggins & Tejani, 2022; Achiam et al., 2023). In fields such as healthcare, law, and radiology, where precision is critical, relying on LLM outputs without factuality guarantees can lead to severe consequences (Katz et al., 2024; Singhal et al., 2023). A key obstacle to reliable deployment is hallucination, where LLMs generate fluent but factually incorrect responses (Farquhar et al., 2024; Zhang et al., 2025; Huang et al., 2025). This motivates us to provide theoretical guarantees on the factual correctness of LLM outputs under realistic deployment conditions.

To this end, we study online LLM factuality using conformal prediction, with the goal of providing theoretical guarantees on the correctness of LLM outputs as they are generated in real time. The key challenge is the online distribution shift caused by evolving user interests and emerging topics. For example, empirical studies on large-scale chat logs demonstrate that user queries exhibit rapid *covariate shifts* across diverse topics (e.g., from coding to creative writing) (Zhao et al., 2024). In addition, the underlying factual knowledge itself is subject to *concept drift*, where the correctness of answers evolves over time (Lazaridou et al., 2021). In this paper, we focus on arbitrary distribution shifts to capture realistic deployment scenarios.

Conformal prediction provides a principled framework for uncertainty quantification by constructing prediction sets with finite-sample coverage guarantees (Vovk et al., 2005; Angelopoulos & Bates, 2022). Recent work has extended conformal prediction to open-ended LLM generation by providing probabilistic guarantees of factual correctness (Quach et al., 2024; Mohri & Hashimoto, 2024; Cherian et al., 2024). However, these methods assume exchangeability in the static setting, where calibration and test data are drawn from the same distribution. In realistic LLM deployments, this assumption is often violated due to non-stationary user queries (Tibshirani et al., 2019; Barber et al., 2023). As a result, these LLM factuality models can provide either indistinguishably large (useless) or deceptively narrow (overconfident) prediction sets, and thus they fail to provide valid

coverage guarantees in online settings.

To bridge this gap, we provide a theoretical analysis of a generic LLM conformal inference framework with a dynamic step size $\gamma_t$ for sequentially adjusting the time-varying target miscoverage parameter $\alpha_t$ at time $t$, inspired by (Gibbs & Candès, 2021). Unlike prior work, our analysis establishes local optimality at time $t$ without relying on asymptotic stationarity assumptions. It also connects the expected miscoverage error to distribution shifts and instantaneous parameter error, thereby resulting in practical guidance for designing the online LLM factuality algorithm. These insights motivate us to propose a novel **P**roactive **A**daptive **C**onformal Inferenc**E** (**PACE**) algorithm by updating the step size $\gamma_t$ based on two complementary signals. The first one is proactive shift detection to estimate the magnitude of distribution shifts, and the second one is reactive error tracking as a proxy of the instantaneous parameter error. In contrast, existing online conformal prediction methods (Bastani et al., 2022; Zaffran et al., 2022; Gibbs & Candès, 2024) rely only on past errors to adjust the step size. When applied to LLM factuality, they ignore the semantic information in the current query $X_t$, which results in delayed adaptation under distribution shifts. By incorporating the semantic distance between the incoming query and the calibration distribution, PACE adjusts the step size to respond more quickly to distribution shifts, thus leading to more stable factuality guarantees. The effectiveness of the proposed PACE algorithm is validated on both controlled synthetic datasets (with abrupt and smooth shifts) and real-world LLM deployment scenarios (including multi-choice question answering and open-ended text generation tasks). Empirical results demonstrate that PACE significantly outperforms advanced baselines: it accelerates coverage recovery by over $2.5\times$ in abrupt shift scenarios and reduces the deviation from target coverage (RMSE) by up to 60% on real-world benchmarks, all while maintaining the highest claim retention rate among adaptive methods.

Our main contributions are summarized as follows:

- We provide the theoretical analysis of a generic LLM conformal inference framework, by explicitly connecting the expected miscoverage error with distribution shifts and instantaneous parameter error.
- Motivated by the theoretical insights, we propose a novel PACE[1] algorithm by unifying reactive error tracking with proactive shift detection.
- Extensive experiments on controlled synthetic datasets and real-world LLM deployment scenarios demonstrate that PACE adapts more quickly to distribution shifts and maintains more stable factuality guarantees compared to baselines.

---

[1]Code is available at the project repository.

## 2. Related Work

**Conformal Prediction for LLMs.** Conformal Prediction (CP) (Vovk et al., 2005; Shafer & Vovk, 2008) constructs prediction sets with finite-sample coverage guarantees. In the context of LLMs, recent works have focused on defining appropriate non-conformity scores for complex language tasks. Kumar et al. (2023) applied CP to multi-choice question answering, while Ravfogel et al. (2023) and Quach et al. (2024) explored conformal generation by calibrating the cumulative probability mass of the token vocabulary. More recently, Conformal Factuality (CF) (Mohri & Hashimoto, 2024; Cherian et al., 2024) has been introduced to guarantee the factuality of open-ended generation via claim decomposition and retrieval-based scoring. However, these methods typically assume a static exchangeable setting. Our work builds upon the scoring mechanisms of CF but extends its calibration procedure to handle the non-stationary nature of online deployment.

**Conformal Prediction Under Distribution Shifts.** Addressing the violation of exchangeability is a core challenge in CP. For covariate shift, Weighted CP (Tibshirani et al., 2019) reweights calibration samples using density ratios, but accurate density-ratio estimation can be unstable in high-dimensional embedding spaces of LLMs (Lin et al., 2025). DS-CP (Lin et al., 2025) further adapts calibration by reweighting samples according to their proximity to the test prompt, but such per-query reweighting can be expensive for real-time streams. Another line of work treats distribution shift as an online calibration problem. Adaptive Conformal Inference (ACI) (Gibbs & Candès, 2021) updates a time-varying parameter using observed miscoverage feedback, and subsequent variants such as AgACI (Zaffran et al., 2022), DtACI (Gibbs & Candès, 2024), and SAOCP (Bhatnagar et al., 2023) improve this update through learning-rate aggregation, dynamic tuning, or strongly adaptive interval experts. However, these approaches are inherently *reactive*, relying exclusively on historical error feedback which leads to delayed adaptation. In contrast, PACE integrates a *proactive* signal derived from input features, enabling rapid adaptation before errors accumulate without the computational overhead of continuous retrieval.

## 3. Preliminaries

### 3.1. Problem Definition

In this paper, we study the problem of online LLM factuality, where the output space can be large and unconstrained. Let $\mathcal{X}$ denote the space of input prompts and $\mathcal{Y}$ the space of generated responses. An LLM is queried sequentially over time, where at each time step $t = 1, 2, \cdots$, it receives an input prompt $X_t \in \mathcal{X}$ and produces a response $\hat{Y}_t \in \mathcal{Y}$ that should be factually correct. In practice, the distribution $P_t$ of the prompt and its corresponding ground-truth response

$(X_t, Y_t) \sim P_t$ can change over time due to evolving user interests and emerging topics. This results in a non-stationary online data stream. Such shifts may be abrupt (e.g., a sudden topic switch) or gradual (e.g., slowly drifting factual knowledge), and their magnitude is generally unknown in advance. Thus, the goal of online LLM factuality is to provide theoretical guarantees on the factuality of LLM outputs under online distribution shifts.

Specifically, we aim to generate a sequence of prediction sets that maintains *marginal coverage* at a user-specified target error rate $\alpha \in (0, 1)$:

$$\mathbb{P}(Y_t \in \hat{C}_t(X_t, \alpha)) \geq 1 - \alpha, \tag{1}$$

where $\hat{C}_t(X_t, \alpha)$ denotes the prediction set constructed at time $t$ for the input prompt $X_t$ given $\alpha$, and the probability is taken over the data-generating distribution $(X_t, Y_t) \sim P_t$. Intuitively, the quantity $\mathbb{P}(Y_t \in \hat{C}_t(X_t, \alpha))$ measures how likely the response generated for $X_t$ is to be factually correct. In the context of LLM factuality, we define the realized miscoverage error $err_t$ based on the task type: for simple question answering (QA), $err_t = \mathbb{I}(Y_t \notin \hat{C}_t)$; for open-ended generation with multiple atomic claims, $err_t = |\mathcal{E}_{\text{fail}}|/|\hat{C}_t|$, representing the proportion of incorrect claims (Mohri & Hashimoto, 2024). For a summary of notations, see Appendix A.1.

### 3.2. Conformal Factuality (CF)

Building on standard conformal prediction (Shafer & Vovk, 2008), Mohri & Hashimoto (2024) introduced Conformal Factuality (CF) to guarantee the correctness of open-ended LLM outputs. Instead of constructing discrete prediction sets, CF employs a nested back-off function $\mathsf{F}_\lambda(\cdot)$ (e.g., retaining only high-confidence atomic claims) and maps the generated response to a validity domain via an information entailment function $\mathsf{E}(\cdot)$. We provide the formal definitions of $\mathsf{E}(\cdot)$ and $\mathsf{F}_\lambda(\cdot)$, along with the detailed scoring mechanism in Appendix A.2.

CF calibrates a strictness threshold $\hat{\lambda}$ on a held-out calibration dataset $\mathcal{D}_{cal}$. Under the assumption of exchangeability between calibration and test data, this procedure guarantees that the generated content entails the ground truth $Y_{test}$ with high probability:

$$\mathbb{P}\left(Y_{test} \in \mathsf{E}\left(\mathsf{F}_{\hat{\lambda}}(X_{test})\right)\right) \geq 1 - \alpha. \tag{2}$$

However, the validity of Eq. (2) relies strictly on the exchangeability assumption. In real-world online deployment, this assumption is often violated due to time-evolving user queries and continuous distribution shifts, leading to coverage failures.

### 3.3. Adaptive Conformal Inference

To handle online distribution shifts, Gibbs & Candès (2021) extended traditional conformal prediction to adaptive conformal inference (ACI) by dynamically adjusting the target error rate $\alpha_t$ at time $t$, instead of keeping it fixed at the user's desired $\alpha$. At each time step $t$, the realized error $err_t$ is observed, and then the parameter $\alpha_t$ is updated:

$$\alpha_{t+1} = \alpha_t + \gamma(\alpha - err_t). \tag{3}$$

The updated $\alpha_{t+1}$ is then used to recompute the threshold $\hat{\lambda}_{t+1}$ for the next step to produce the prediction set for $X_{t+1}$. This mechanism allows the system to tighten (increase $\lambda$) or loosen (decrease $\lambda$) the constraints dynamically as the difficulty of the online stream changes. However, the fixed step size $\gamma$ in Eq. (3) may lead to suboptimal adaptation under rapidly changing online distributions.

## 4. Theoretical Analysis

### 4.1. A Generic Framework

We start by introducing a generic adaptive conformal inference framework for handling online distribution shifts in input prompts for language models.

Following (Gibbs & Candès, 2021), the mapping function $\alpha \mapsto \mathbb{P}_{(X,Y)\sim P_t}(Y \notin \hat{C}(X, \alpha))$ is assumed to be continuous. Moreover, since this mapping is non-decreasing with $\mathbb{P}_{(X,Y)\sim P_t}(Y \notin \hat{C}(X, 0)) = 0$ and $\mathbb{P}_{(X,Y)\sim P_t}(Y \notin \hat{C}(X, 1)) = 1$, there exists a latent time-varying optimal parameter $\alpha_t^*$ that achieves the target error rate under the instantaneous distribution

$$\mathbb{P}_{(X,Y)\sim P_t}(Y \notin \hat{C}(X, \alpha_t^*)) = \alpha \tag{4}$$

Since the data-generation distribution $P_t$ may shift arbitrarily over time, the optimal parameter $\alpha_t^*$ follows a non-stationary trajectory. Our objective is to learn a sequence $\{\alpha_t\}_{t \geq 1}$ to track this evolving target in an online manner. To this end, we introduce the following update rule:

$$\alpha_{t+1} = \alpha_t + \gamma_t(\mathcal{S}_t) \cdot (\alpha - \text{err}_t), \tag{5}$$

where $\gamma_t(\mathcal{S}_t)$ is a dynamic step-size function dependent on a state context $\mathcal{S}_t$ available at time $t$.

This framework can be seen as a natural generalization of traditional conformal prediction. If $\gamma_t(\mathcal{S}_t) = 0$, our framework degenerates to standard conformal inference where the value $\alpha_t$ is held constant across all timestamps. When $\gamma_t(\mathcal{S}_t) = \gamma$ for some $\gamma > 0$, it reduces to traditional adaptive conformal inference (ACI) (Gibbs & Candès, 2021) with a fixed step size to handle online distribution shifts. Although the limitations of the fixed step size $\gamma$ on the update dynamics have been empirically explored in previous

work (Zaffran et al., 2022; Gibbs & Candès, 2024), a principled theoretical analysis to guide the design of the dynamic step-size function $\gamma_t(\mathcal{S}_t)$ is still lacking. To bridge the gap, we provide a theoretical analysis of our generic framework in the following.

## 4.2. Asymptotic Coverage Guarantees

We analyze the asymptotic coverage properties of our framework based on the update rule in Eq. (5). In particular, we show that the proposed framework can achieve a target coverage level $\alpha \in (0, 1)$ in the long run under online distribution shifts. For notation simplicity, we write $\gamma_t := \gamma_t(\mathcal{S}_t)$ for the step size at time $t$ below.

**Lemma 4.1** (Boundedness of $\alpha_t$). *Assume the step-size function $\gamma_t$ is bounded with $\gamma_t \in [\gamma_{\min}, \gamma_{\max}]$ where $\gamma_{\min} \geq 0$. Then, with probability one, the sequence $\{\alpha_t\}_{t \geq 1}$ is bounded within the interval $[-\gamma_{\max}, 1 + \gamma_{\max}]$.*

*Proof.* See Appendix B.1. □

Based on the boundedness of $\alpha_t$ for all $t$, the following theorem shows the asymptotic coverage guarantees of our framework as the time horizon $T$ goes to infinity.

**Theorem 4.2** (Weighted Long-Term Convergence). *Under the assumptions of Lemma 4.1 (i.e., bounded step-sizes), let $W_T = \sum_{t=1}^{T} \gamma_t$ be the cumulative adaptive weight. If $W_T \to \infty$ as $T \to \infty$, then with probability one:*

$$\lim_{T \to \infty} \left| \frac{\sum_{t=1}^{T} \gamma_t(err_t - \alpha)}{W_T} \right| = 0. \qquad (6)$$

*Proof.* See Appendix B.2. □

*Remark* 4.3 (Accelerated Adaptation). Theorem 4.2 implies the superior efficiency of our generic framework compared to fixed-step ACI (Gibbs & Candès, 2021). The convergence rate of our framework is determined by $O(1/W_T)$. In Standard ACI with fixed step size $\gamma_t = \gamma$, it holds that $W_T = T\gamma$. The system forgets historical errors linearly. In contrast, the adaptive step size $\gamma_t$ in our framework can spike towards $\gamma_{max}$ during distribution shifts (discussed in Subsection 4.3). Consequently, the cumulative weight $W_T$ grows significantly faster than linear time during volatile periods. This mechanism allows our framework to "forget" obsolete historical statistics and re-converge to the target $\alpha$ at an accelerated rate when it matters most, explaining the rapid coverage convergence. Appendix C.6 provides an empirical illustration of this accelerated convergence behavior on real-world data stream.

## 4.3. Optimality of the Update Rule

We now provide a fine-grained theoretical analysis of the update rule in Eq. (5) to characterize conditions under which

the step size $\gamma_t$ is optimal.

Following (Gibbs & Candès, 2021), we model the online data generation process using a Hidden Markov Model. Let $\{A_t\}_{t \geq 1} \subseteq \mathcal{A}$ denote the latent environment state. Conditional on $A_t$, the observation $(X_t, Y_t)$ is drawn from a state-dependent distribution $P_{A_t}$. We define the time-varying *miscoverage function* $M_t(\cdot)$ as the conditional probability that the prediction set fails to cover the ground-truth $Y_t$ given the environment state $A_t$:

$$M_t(\alpha) := \mathbb{P}(Y_t \notin \hat{C}(X_t, \alpha) \mid A_t). \qquad (7)$$

with respect to $(X_t, Y_t) \sim P_{A_t}$. The miscoverage function $M_t(\cdot)$ is non-decreasing with $M_t(0) = 0$ and $M_t(1) = 1$, and then the *ideal conformal parameter* $\alpha_t^*$ can be defined as:

$$\alpha_t^* := \sup\{\beta \in [0, 1] \mid M_t(\beta) \leq \alpha\} \qquad (8)$$

Unlike Eq. (4) that may admit multiple optimal values for $\alpha_t^*$, this definition yields a unique ideal conformal parameter. This allows us to characterize the distribution shift between consecutive steps by $\Delta_t = \alpha_{t+1}^* - \alpha_t^*$. In the absence of distribution shifts during online LLM inference, it holds $\Delta_t = 0$ for all $t$.

In the online LLM factuality setting, the objective of the update rule in Eq. (5) is to choose the step size $\gamma_t$ by minimizing the expected squared miscoverage error:

$$\min_{\gamma_t} \mathcal{J}(\gamma_t) := \mathbb{E}[(M_{t+1}(\alpha_{t+1}) - \alpha)^2 \mid \mathcal{F}_t]. \qquad (9)$$

Here, $\mathcal{F}_t = \{(\alpha_1, A_1), \ldots, (\alpha_t, A_t)\}$ denotes the *observed history* of the system up to time $t$. However, direct optimization of $\mathcal{J}$ is intractable because the explicit form of $M_t(\cdot)$ is unknown. To solve this problem, we consider the *parameter tracking error* $\Phi_{t+1} = (\alpha_{t+1} - \alpha_{t+1}^*)^2$, which captures the deviation of the conformal parameter from its ideal value. The intuition is that if $M_t(\cdot)$ is assumed to be $L$-Lipschitz, the tracking error provides an upper bound on the miscoverage error, i.e., $(M_t(\alpha_{t+1}) - \alpha)^2 \leq L^2(\alpha_{t+1} - \alpha_{t+1}^*)^2$. Thus, minimizing the *parameter tracking error* $\Phi_{t+1}$ provides a tractable surrogate for the original objective $\mathcal{J}(\gamma_t)$:

$$\min_{\gamma_t} \mathbb{E}[\Phi_{t+1} | \mathcal{F}_t] = \mathbb{E}[(\alpha_{t+1} - \alpha_{t+1}^*)^2 | \mathcal{F}_t]. \qquad (10)$$

**Assumption 4.4.** We consider the following assumption: The miscoverage function $M_t(\cdot)$ is assumed to be $L$-Lipschitz continuous and non-decreasing. Specifically, there exist constants $L \geq \rho \geq 0$ such that for any $\alpha, \alpha' \in [0, 1]$:

$$\rho(\alpha - \alpha')^2 \leq (M_t(\alpha) - M_t(\alpha'))(\alpha - \alpha') \leq L(\alpha - \alpha')^2.$$

Under Assumption 4.4, we derive an optimal adaptive step size as follows.

**Theorem 4.5** (Optimal Dynamic Step-Size). *Under Assumption 4.4, an upper bound of the expected tracking error in Eq. (10) is minimized by an unconstrained step size*

$$\gamma_t^* \propto \rho \cdot \Phi_t + \mathbb{E}_t\left[(\alpha - \mathrm{err}_t)\Delta_t\right], \quad (11)$$

*where $\Phi_t = (\alpha_t - \alpha_t^*)^2$ represents the instantaneous parameter error, and the second term is a signed drift-feedback term involving the latent shift $\Delta_t = \alpha_{t+1}^* - \alpha_t^*$.*

*Proof.* See Appendix B.3 for details. □

*Remark* 4.6. We would like to point out that compared to traditional ACI (Gibbs & Candès, 2021), Theorem 4.5 provides several new insights. First, it establishes local optimality at every time step $t$, without relying on asymptotic stationary assumptions. Second, it provides a rigorous theoretical foundation for our practical update rule in Eq. (5) (will be discussed in Section 5).

## 5. Methodology

Motivated by the theoretical results in Section 4, we propose **PACE** (Proactive Adaptive Conformal Estimation), which instantiates the update rule in Eq. (5) with a computable dynamic step size. Theorem 4.5 indicates that the adaptive step size should scale with two latent quantities: the instantaneous parameter error $\Phi_t$ and a signed drift-feedback term involving $\Delta_t$. Since both $\alpha_t^*$ and $\Delta_t$ are unobserved in online deployment, PACE replaces them with two computable proxies: $S_{\mathrm{error}}$ estimates the current calibration mismatch, while $S_{\mathrm{shift}}$ estimates the magnitude of the incoming input-distribution shift. The direction of the conformal update is still determined by the feedback term $\alpha - \mathrm{err}_t$ in Eq. (5).

### 5.1. Constructing the Dual Signals

#### 5.1.1. SIGNAL 1: PROACTIVE SHIFT DETECTION.

The latent parameter shift $\Delta_t = \alpha_{t+1}^* - \alpha_t^*$ is primarily induced by deviations in the input distribution. Intuitively, if the current prompt $X_t$ deviates significantly from the calibration distribution (i.e., is Out-Of-Distribution), the model's prediction uncertainty is likely to change, necessitating an adjustment in $\alpha_t^*$. Therefore, we design $S_{\mathrm{shift}}$ to quantify the magnitude of this input distribution shift as an indicator for $\Delta_t$. To estimate this shift, we leverage the feature embeddings $f(X_t)$ of the input prompt. Since simple Euclidean metrics fail due to the anisotropic nature of pre-trained LLM embeddings (Ethayarajh, 2019), we employ the **Relative Mahalanobis Distance (RMD)** (Ren et al., 2023).

RMD isolates meaningful semantic shifts by contrasting task-specific deviations against general background noise. We maintain two online Gaussian estimators: $(\boldsymbol{\mu}_{in}, \boldsymbol{\Sigma}_{in})$ derived from the calibration set (In-Distribution), and

$(\boldsymbol{\mu}_{bg}, \boldsymbol{\Sigma}_{bg})$ computed on a generic corpus like Wikipedia (Background). The relative shift score $D(X_t)$ is defined as the difference between their respective Mahalanobis Distances (MD):

$$D(X_t) = \mathrm{MD}(X_t; \boldsymbol{\mu}_{in}, \boldsymbol{\Sigma}_{in}) - \mathrm{MD}(X_t; \boldsymbol{\mu}_{bg}, \boldsymbol{\Sigma}_{bg}), \quad (12)$$

where $\mathrm{MD}(x; \boldsymbol{\mu}, \boldsymbol{\Sigma}) = \sqrt{(f(x) - \boldsymbol{\mu})^\top \boldsymbol{\Sigma}^{-1}(f(x) - \boldsymbol{\mu})}$. Intuitively, subtracting the background distance acts as a background cancellation mechanism, ensuring that high scores reflect true domain shifts rather than generic linguistic variability. Finally, we map this score to a normalized probability $S_{\mathrm{shift}} \in [0, 1]$ via the sigmoid function: $S_{\mathrm{shift}} = (1 + e^{-D(X_t)})^{-1}$.

#### 5.1.2. SIGNAL 2: REACTIVE ERROR TRACKING.

The parameter error $\Phi_t = (\alpha_t - \alpha_t^*)^2$ reflects the misalignment between the current and optimal control parameters. We approximate this latent error using the deviation of the realized error rate from the target $\alpha$ over a sliding window of size $n$:

$$S_{\mathrm{error}} = |\hat{\pi}_t - \alpha|, \quad \text{where } \hat{\pi}_t = \frac{1}{n} \sum_{i=t-n+1}^{t} \mathrm{err}_i. \quad (13)$$

The windowed average $\hat{\pi}_t$ acts as a low-pass filter. Since the instantaneous feedback $\mathrm{err}_t$ is highly stochastic (e.g., binary flips or aleatoric uncertainty), using it directly would cause severe parameter oscillation.

### 5.2. The PACE Algorithm

Based on the proactive signal $S_{\mathrm{shift}}$ and the reactive signal $S_{\mathrm{error}}$ derived above, we formulate the dynamic step size $\gamma_t$ as follows:

$$\gamma_t(\mathcal{S}_t) = c_1 \cdot S_{\mathrm{error}} + c_2 \cdot S_{\mathrm{shift}}, \quad (14)$$

where $c_1, c_2 \in [0, 1]$ are nonnegative scaling weights for the two bounded proxies. These weights control the magnitude of the adaptive step size rather than the direction of the update. The update direction is determined by the feedback term $\alpha - \mathrm{err}_t$ in Eq. (5). In this sense, $S_{\mathrm{error}}$ provides a computable proxy for the parameter-misalignment component $\Phi_t$, while $S_{\mathrm{shift}}$ provides a computable proxy for the magnitude of the latent input-distribution drift suggested by Theorem 4.5.

Algorithm 1 outlines the PACE workflow, designed as a dynamic safety wrapper for online LLM factuality. The process iterates through three key phases at time $t$:

(i) **Calibration:** The current target $\alpha_t$ is mapped to a rejection threshold $\lambda_t$ by computing the $(1-\alpha_t)$-quantile of non-conformity scores from the calibration set $\mathcal{D}_{cal}$.

---

**Algorithm 1** PACE for Online LLM Factuality

---

1: **Input:** Target error $\alpha$, Calibration set $\mathcal{D}_{cal}$, Hyperparams $c_1, c_2 \in [0, 1]$, Window size $n$.
2: **Initialize:** $\alpha_1 \leftarrow \alpha$.
3: **Observe initial query:** $X_1$
4: **for** $t = 1, 2, \ldots$ **do**
5:    **Step 1: Calibration & Prediction**
6:    $\lambda_t \leftarrow \text{Quantile}(1 - \alpha_t; \mathcal{D}_{cal})$
7:    $Y^0 \leftarrow \text{LLM}(X_t)$
8:    $\mathcal{A}_{all} \leftarrow \text{ExtractClaims}(Y^0)$
9:    $\hat{Y}_t \leftarrow \text{Filter}(\{k \in \mathcal{A}_{all} \mid \text{Score}(k) \leq \lambda_t\})$
10:   **Output** $\hat{Y}_t$
11:   **Step 2: Feedback & Adaptation**
12:   Observe ground truth $Y_t^*$
13:   $\mathcal{E}_{fail} \leftarrow \{k \in \hat{Y}_t \mid \text{Verify}\,(k, Y_t^*) \text{ is False}\}$
14:   **if** $|\hat{Y}_t| > 0$ **then**
15:      $\text{err}_t \leftarrow |\mathcal{E}_{fail}|/|\hat{Y}_t|$
16:   **else**
17:      $\text{err}_t \leftarrow 0$ {No claims, no error}
18:   **end if**
19:   $\hat{\pi}_t \leftarrow \frac{1}{n} \sum_{i=t-n+1}^{t} \text{err}_i$
20:   **Step 3: Adaptation for Time** $t + 1$
21:   Observe next query $X_{t+1}$
22:     $\gamma_t \leftarrow c_1 \cdot S_{error} + c_2 \cdot S_{shift}$
23:     $\alpha_{t+1} \leftarrow \alpha_t + \gamma_t(\alpha - \text{err}_t)$
24: **end for**

---

(ii) **Inference & Filtering:** The LLM generates a raw response to $X_t$, which is decomposed into atomic claims. Only claims with uncertainty scores below $\lambda_t$ are retained in the final output $\hat{Y}_t$.

(iii) **Adaptation:** The system observes the realized error $\text{err}_t$ (via an external verifier) and computes the semantic shift $S_{shift}$ of the next query. These signals are used to update the target error rate $\alpha_{t+1}$ via Eq. (5) and (14), closing the feedback loop.

# 6. Synthetic Data Simulation

In this section, we validate the theoretical properties of PACE in a controlled environment where the exact ground-truth parameters are known.

## 6.1. Experimental Setup

We validate PACE on a simulated scalar data stream $Y_t \sim \mathcal{N}(\mu_t, 1)$ over $T = 6000$ steps. The mean $\mu_t$ varies to model two distinct distribution shifts: (1) **Abrupt Shift**, where $\mu_t$ acts as a piecewise constant function with sudden discrete jumps; and (2) **Smooth Shift**, where $\mu_t$ evolves via a momentum-based random walk. The trajectories of $\mu_t$ under abrupt and smooth shifts are shown in Figure 1a and Figure 1b, respectively. To compute the proactive signal $S_{shift}$, we measure the deviation of the current observation from a local background estimated via a sliding window.

**Baselines.** We consider the following baseline methods: (1) Standard Conformal Prediction (SCP) with fixed $\alpha = 0.1$.

(2) Adaptive Conformal Inference (ACI) (Gibbs & Candès, 2021) with fixed step sizes. We carefully select step sizes approximating the lower and upper bounds of PACE's adaptive $\gamma$ range (e.g., $\gamma \in \{0.01, 0.2\}$ for Abrupt Shift Scenario and $\gamma \in \{0.01, 0.1\}$ for Smooth Shift Scenario), representing conservative and aggressive strategies, respectively. This allows us to evaluate whether PACE can dynamically interpolate between these regimes. (3) DtACI (Gibbs & Candès, 2024), an advanced adaptive method that outperforms previous adaptive methods by dynamically tuning the step size $\gamma_t$ via an exponential re-weighting scheme (i.e., expert aggregation). We additionally compare with SAOCP (Bhatnagar et al., 2023), another strongly adaptive online conformal method, in Appendix C.8.

**Evaluation Metrics.** (1) **Coverage**: The empirical coverage rate is measured as $\frac{1}{T} \sum_{t=1}^{T} \mathbb{I}\{Y_t \in \hat{C}_t\}$. A valid calibrator should have this rate converge asymptotically to the target level $1 - \alpha$. (2) **RMSE (Root Mean Squared Error)**: We measure the error gap with the target error rate $\alpha$ by calculating the deviation between the local realized error rate $\hat{\pi}_t$ (estimated via a sliding window) and the target $\alpha$: $\text{RMSE} = \sqrt{\frac{1}{T} \sum_{t=1}^{T} (\hat{\pi}_t - \alpha)^2}$. A lower RMSE indicates that the algorithm tightly tracks the target error rate with minimal oscillation. (3) **Recovery Time** ($T_{rec}$): To assess reactivity to the abrupt shifts, we define $T_{rec}$ as the number of time steps required for the rolling local error rate to return to the valid range $[\alpha - \epsilon, \alpha + \epsilon]$ ($\epsilon = 0.05$) after a distribution abrupt shift occurs.

Detailed generation processes and hyperparameter settings are provided in Appendix C.1.

## 6.2. Results and Analysis

We validate the effectiveness of PACE using both qualitative visualizations and quantitative metrics. Based on the parameter sensitivity analysis provided in Appendix C.4.1, we select representative configurations for our experiments: we set $c_1 = 0.7, c_2 = 0.3$ for the Abrupt Shift Scenario and $c_1 = 0.5, c_2 = 0.5$ for the Smooth Shift Scenario. Figure 1c and Figure 1d show the trajectory of the conformal parameter $\alpha_t$ tracked by each algorithm against the ground truth $\alpha_t^*$ (black line), directly visualizing the parameter tracking capability. Figure 1e and Figure 1f depict the dynamics of the realized local error rate (calculated via a sliding window of size $W = 100$). Quantitative results are summarized in Tables 1 and 2, respectively. We provide parameter sensitivity analysis in Appendix C.4.1.

In the Abrupt Shift scenario, ACI faces an inherent trade-off: a small fixed $\gamma$ (e.g., 0.01) results in significant lag, failing to recover coverage for hundreds of steps after a jump although its average coverage is close to the target level $1 - \alpha$. Conversely, a large $\gamma$ (e.g., 0.2) induces exces-

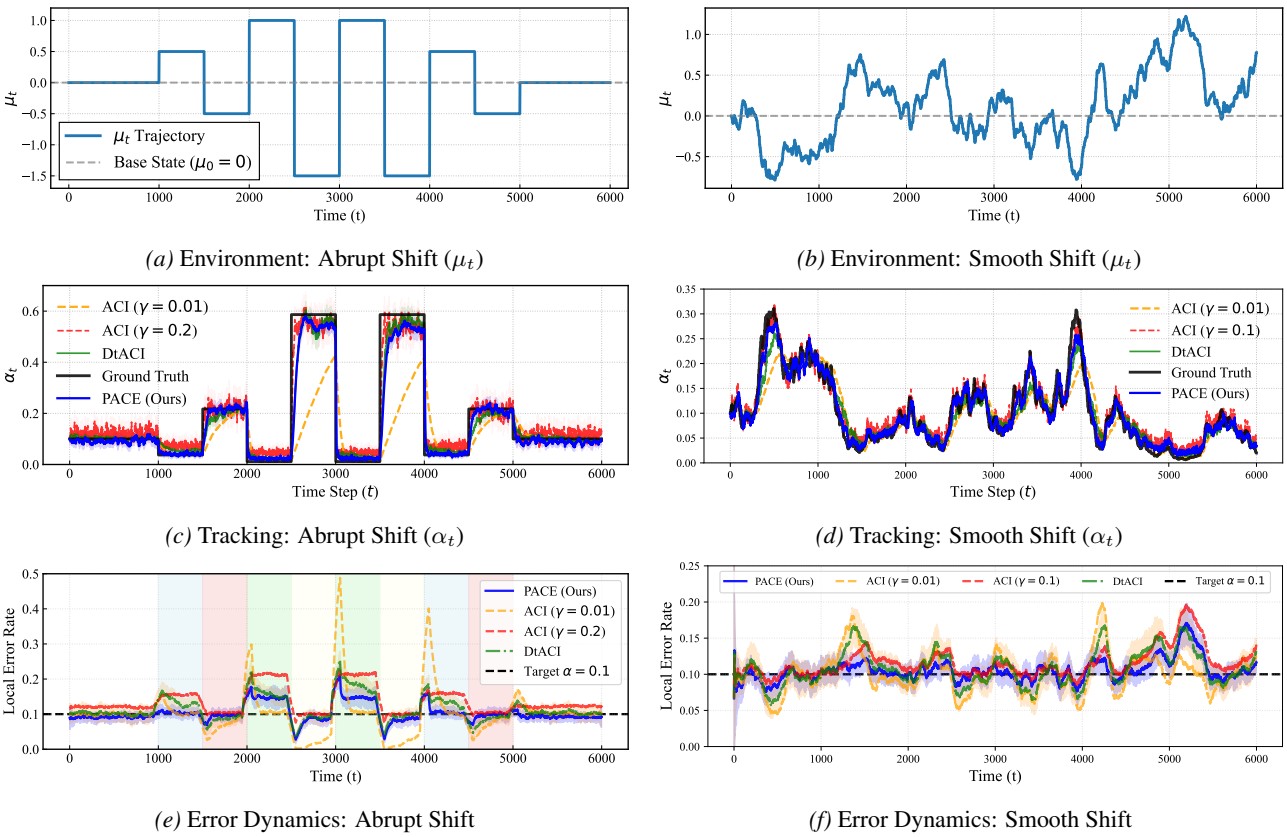

*Figure 1.* **Evaluation on synthetic data streams. Top Row:** Visualization of the trajectories of mean parameter $\mu_t$ in (a) Abrupt Shift and (b) Smooth Shift scenarios. **Middle Row:** Conformal parameter $\alpha_t$ tracking trajectories. While reactive baselines (dashed lines) suffer from lag or oscillation, PACE (blue solid line) leverages the proactive signal to track the optimal $\alpha_t^*$ (black) with near-instantaneous adaptation. **Bottom Row:** Dynamics of realized local error rates. PACE effectively anticipates shifts, significantly dampening error spikes in abrupt scenarios (e) and maintaining the tightest stability around the target $\alpha = 0.1$ in smooth scenarios (f).

*Table 1.* Results of conformal prediction under **Abrupt** shifts

| Method | Coverage | RMSE | Recovery |
|---|---|---|---|
| SCP | $0.8598 \pm 0.0038$ | $0.1321 \pm 0.0044$ | 342.48 |
| ACI ($\gamma = 0.01$) | $\mathbf{0.8992 \pm 0.0005}$ | $0.0789 \pm 0.0019$ | 135.06 |
| ACI ($\gamma = 0.2$) | $0.8642 \pm 0.0015$ | $0.0542 \pm 0.0022$ | 158.27 |
| DtACI | $0.8862 \pm 0.0019$ | $0.0461 \pm 0.0024$ | 120.66 |
| PACE | $0.8958 \pm 0.0019$ | $\mathbf{0.0349 \pm 0.0023}$ | **46.06** |

*Table 2.* Results of conformal prediction under **Smooth** shifts

| Method | Coverage | RMSE |
|---|---|---|
| SCP | $0.8592 \pm 0.0035$ | $0.1067 \pm 0.0050$ |
| ACI ($\gamma = 0.01$) | $\mathbf{0.8984 \pm 0.0003}$ | $0.0355 \pm 0.0022$ |
| ACI ($\gamma = 0.2$) | $0.8840 \pm 0.0009$ | $0.0288 \pm 0.0018$ |
| DtACI | $0.8917 \pm 0.0016$ | $0.0325 \pm 0.0022$ |
| PACE | $0.8947 \pm 0.0013$ | $\mathbf{0.0238 \pm 0.0018}$ |

sive oscillation during stable periods. DtACI improves over ACI by dynamically adjusting $\gamma_t$ using past errors, but it is still *reactive*, as it only updates after observing coverage failures. This is visibly evident in Figure 1e, where DtACI (green line) slowly adjusts $\alpha_t$ over hundreds of steps during large distribution shifts (e.g., $t = 1000$ $t = 1500$ and $t = 4000$). Consequently, its Recovery Time (120.66) is

nearly $3\times$ slower than PACE. In contrast, PACE leverages the input-driven signal $S_{\text{shift}}$ to directly detect distribution changes in the input embeddings without waiting for errors to accumulate. As shown in Figure 1e, PACE (blue line) closely tracks the ground-truth $\alpha_t^*$ (black line). For distribution shifts at any time, PACE adjusts $\alpha_t$ almost instantaneously. This explains its superior Recovery Time (46.06) and Tracking RMSE (0.0349) in Table 1.

In the Smooth Shift Scenario, where distribution shifts are gradual and subtle, PACE achieves the lowest Tracking RMSE. This indicates that PACE effectively adjusts to subtle shifts, whereas DtACI struggles to find the optimal adaptation rate in the absence of persistent error signals. As shown in Figure 1d and Figure 1f, PACE achieves near-perfect tracking of the ground-truth $\alpha_t^*$ (black line) with minimal variance. Appendices C.8.2 and C.8.3 provide additional robustness checks under complex mixture shifts and time-varying non-Gaussian Beta shifts, where PACE maintains the lowest tracking RMSE and smallest worst violation. Appendix C.9 further analyzes prediction set size dynamics on MMLU, showing that PACE expands and contracts sets

Table 3. Results on the MMLU dataset (QA Task).

| Method | Factuality | RMSE | Recovery | Set Size |
|---|---|---|---|---|
| SCP | $88.07 \pm 3.40$ | $0.0360 \pm 0.0133$ | 149.0 | $\mathbf{1.00 \pm 0.08}$ |
| ACI($\gamma = 0.01$) | $\mathbf{89.91 \pm 0.03}$ | $0.0155 \pm 0.0039$ | 24.9 | $1.08 \pm 0.05$ |
| ACI($\gamma = 0.1$) | $89.00 \pm 0.52$ | $0.0120 \pm 0.0042$ | **1.8** | $1.27 \pm 0.08$ |
| DtACI | $89.73 \pm 0.29$ | $0.0175 \pm 0.0032$ | 24.6 | $1.07 \pm 0.04$ |
| PACE | $89.85 \pm 0.37$ | $\mathbf{0.0068 \pm 0.0024}$ | 3.0 | $1.17 \pm 0.06$ |

Table 4. Results on the WikiData dataset (Open-ended Generation).

| Method | Factuality | RMSE | Recovery | Claims Retained |
|---|---|---|---|---|
| CF | $78.15 \pm 16.54$ | $0.1664 \pm 0.0030$ | 140.25 | $\mathbf{78.82 \pm 19.92}$ |
| ACI($\gamma = 0.01$) | $79.75 \pm 16.49$ | $\mathbf{0.1649 \pm 0.0028}$ | 89.98 | $71.94 \pm 21.35$ |
| ACI($\gamma = 0.2$) | $79.96 \pm 16.78$ | $0.1678 \pm 0.0031$ | 31.98 | $72.17 \pm 24.98$ |
| DtACI | $79.91 \pm 16.56$ | $0.1656 \pm 0.0027$ | 59.66 | $71.97 \pm 23.14$ |
| PACE | $\mathbf{79.96 \pm 16.70}$ | $0.1670 \pm 0.0029$ | 39.57 | $72.27 \pm 24.48$ |

in proportion to stream difficulty. Appendix C.10 reports results with Llama-2-70B as the base model, where PACE again achieves the lowest tracking RMSE while maintaining factuality close to the target level.

# 7. Real-World Experiments

## 7.1. Settings

**Datasets and Shift Simulation.** We evaluate PACE on two diverse widely used benchmarks designed to simulate non-stationary online streams. (1) **MMLU** (Hendrycks et al., 2021): a comprehensive benchmark evaluating knowledge and reasoning across 57 subjects. To simulate domain shifts, we construct a sequential stream by concatenating questions from distinct subjects (e.g., Abstract Algebra → Clinical Knowledge). (2) **WikiData** (Cherian et al., 2024): an open-ended biography generation dataset constructed in (Cherian et al., 2024) by sampling entity names from Wikipedia. In our experiments, we follow FActScore (Min et al., 2023) to create the dataset including 3500 entities by querying frontier LLM GPT-5.2 to generate biographies and annotate them and using GPT-4o-mini to parse the response into claims. We use high-resource entities (top 25% popularity) for calibration and lower-resource entities for the test stream, introducing a natural distribution shift where the model's hallucination rate increases over time. Details on stream construction are provided in Appendix C.2.

**Experimental Setup.** We utilize QWEN-72B for MMLU and GPT-5.2 for WikiData. For scoring, we employ the Least Ambiguous Classifier (LAC) score (Lin et al., 2025) for QA, confidence and frequency scoring (Mohri & Hashimoto, 2024) for generation. Ground truth correctness is verified using an automated LLM oracle following (Min et al., 2023).

**Baselines.** We compare our PACE method against strong conformal inference baselines. For multi-choice QA on MMLU, the prediction set consists of the candidate labels. The baselines include standard conformal prediction (SCP), adaptive conformal inference (ACI) (Gibbs &

Candès, 2021), and dynamically-tuned adaptive conformal inference (DtACI) (Gibbs & Candès, 2024). For open-ended text generation on WikiData, the prediction set is defined as the set of atomic claims retained in the generated response. Here, we use conformal factuality (CF) (Mohri & Hashimoto, 2024) as the baseline. Besides, we extend ACI and DtACI to the CF setting to evaluate their effectiveness in controlling factuality for LLM-generated responses.

**Evaluation Metrics.** We adopt the metrics defined in Section 6.1, with a task-specific adaptation for **RMSE** on WikiData. As mentioned in Section 3.1, $err_t \in [0, 1]$ (proportion of unsupported claims) is not binary, so we calculate RMSE directly using the instantaneous error $err_t$ to rigorously capture sample-level tracking fidelity instead of the sliding-window estimate. We target an error rate of $\alpha = 0.1$ for MMLU and $\alpha = 0.2$ for WikiData. Additionally, we introduce more utility metrics: (1) **Set Size (MMLU):** The average number of candidate options in the prediction set $\hat{C}_t$. Smaller sets imply higher precision. (2) **Factuality & Claims Retained (WikiData):** To evaluate open-ended generation, we report: ① **Factuality (%):** The proportion of atomic claims in the final response supported by Wikipedia (equivalent to $1 - $ average $err_t$). ② **Claims Retained (%):** The percentage of atomic claims preserved after conformal filtering relative to the original raw generation.

## 7.2. Results and Analysis

**MMLU.** Guided by the parameter sensitivity and ablation analysis detailed in Appendix C.4.2, we configure PACE with $c_1 = 0.6$ and $c_2 = 0.4$ for this experiment. As summarized in Table 3, the static baseline SCP yields the smallest prediction set size (1.00) but fails to meet the coverage target ($88.07\% < 90\%$) with the highest RMSE (0.0175). This indicates that the fixed threshold derived from calibration is too restrictive for the shifting test stream, causing the model to under-cover the ground truth during difficult queries. Among adaptive methods, PACE demonstrates superior tracking fidelity, achieving an RMSE of 0.0068, a reduction of approximately 60% compared to the strongest adaptive baseline DtACI (0.0175). This significant performance gap highlights the critical advantage of our proactive mechanism. Reactive baselines inherently suffer from *lag*: they must accumulate a history of coverage violations to "diagnose" that the current step size is insufficient. As shown in Figure 2, while reactive baselines show sharp spikes in the realized error rate during topic transitions, PACE maintains a stable error trajectory. Our method preemptively tightens constraints, achieving near-instantaneous recovery ($T_{rec} \approx 3.0$) without waiting for coverage violations to accumulate.

**WikiData.** Using a balanced configuration ($c_1 = c_2 = 0.5$, supported by the analysis in Appendix C.4.2), as shown in

Table 4, the static baseline CF achieves the highest Claim Retention rate but violates the factuality guarantee, confirming that a fixed threshold is too permissive for low-resource entities prone to hallucination. Among the adaptive methods, PACE achieves comparable RMSE to the baselines, indicating that our method maintains competitive tracking stability. However, this global RMSE metric masks critical trade-offs made by the baselines. Conservative strategies such as ACI ($\gamma = 0.01$) and DtACI, achieve slightly lower RMSE by ignoring high-frequency fluctuations but fail to adapt to genuine shifts, resulting in prolonged under-coverage with recovery times of 89.98 and 59.66 steps, respectively. Conversely, aggressive baselines (ACI $\gamma = 0.2$) improve recovery speed but degrade stability, exhibiting the highest RMSE and risking over-reaction to aleatoric noise. In contrast, PACE effectively balances these competing objectives. It ensures safety by meeting a near-$80\%$ factuality target and achieves rapid recovery comparable to aggressive strategies. Crucially, PACE yields the highest Claim Retention rate among all valid adaptive methods. This demonstrates that PACE's interventions are precise, maximizing utility without compromising safety. We also validated Remark 4.3 (Accelerated Adaptation) in Appendix C.6 and conducted an extensive sweep of $\gamma \in [0.01, 1.0]$ in Appendix C.5. The results show that ACI is highly sensitive to hyperparameter selection: some configurations perform very poorly on some metrics, even though the best-performing fixed $\gamma$ can't simultaneously minimize tracking error and maximize claim retention. Notably, PACE's adaptive mechanism achieves a Claim Retention rate that exceeds even the best-performing fixed ACI configuration ($\gamma = 0.2$, $72.17\%$), effectively acting as an optimal, dynamic step-size selector. We further provide an online latency analysis in Appendix C.7, showing that PACE adds only negligible algorithmic overhead compared with the shared LLM generation, decomposition, and factuality scoring pipeline.

## 8. Conclusion

In this paper, we have proposed PACE, a novel conformal inference framework for online LLM factuality. PACE leverages a proactive signal derived from the input semantics to dynamically adjust the conformal prediction set, enabling faster adaptation before errors accumulate. We have shown that PACE achieves state-of-the-art performance on a range of tasks while maintaining computational efficiency.

There are several potential limitations for the proposed PACE framework. First, for open-ended generation, our PACE framework follows the Conformal Factuality pipeline and evaluates factuality at the atomic-claim level. This assumes that claim-level errors can be aggregated locally, which may be insufficient for multi-step reasoning tasks where errors propagate across dependent claims. Second,

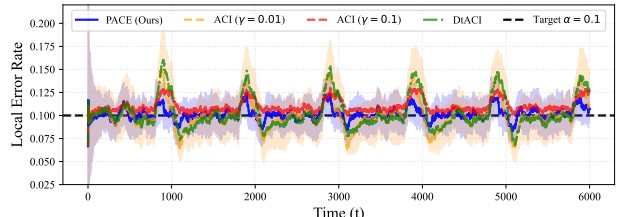

*(a)* Local Error Rate Dynamics on MMLU.

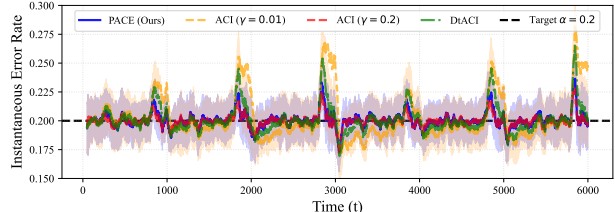

*(b)* Local Error Rate Dynamics on WikiData.

*Figure 2.* Error rate tracking on real-world LLM tasks (a) MMLU and (b) WikiData.

PACE is designed for knowledge-intensive text generation and question answering. Extending it to multimodal settings would require factuality scores that capture visual grounding and cross-modal consistency. Third, as with ACI-based online conformal methods, our coverage result is asymptotic. Deriving non-asymptotic guarantees under arbitrary distribution shifts remains an important direction for future work.

## Acknowledgements

This work is supported by Michigan State University and an Amazon Research Award. The views and conclusions are those of the authors and should not be interpreted as representing the official policies of the funding agencies or the government.

## Impact Statement

This paper studies uncertainty quantification and factuality for large language models in online settings. By adapting conformal prediction under distribution shifts, the proposed method may help improve the reliability of deployed language model systems and reduce overconfident factual errors. However, conformal guarantees depend on the quality of the underlying model, scoring function, feedback signal, and deployment environment. Therefore, this method should be used as part of a broader evaluation and safety pipeline, rather than as a substitute for human oversight in high-stakes applications.

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

# Appendix

## A. Preliminaries Details

### A.1. Notations

*Table 5.* Summary of main notations used in this paper.

| Notation | Description | Notation | Description |
|---|---|---|---|
| $\mathcal{X}, \mathcal{Y}$ | Input prompt space and output response space | $A_t$ | Latent environment state in HMM |
| $X_t, Y_t$ | Input prompt and ground-truth response at time $t$ | $P_{A_t}$ | Data distribution conditioned on state $A_t$ |
| $\alpha$ | Target error rate (e.g., $\alpha = 0.1$) | $M_t(\alpha)$ | Miscoverage function $\mathbb{P}(Y \notin \hat{C}(X_t, \alpha) \mid A_t)$ |
| $\hat{C}(X, \alpha)$ | Conformal prediction set given input $X$ and $\alpha$ | $\alpha_t$ | Adaptive conformal parameter at time $t$ |
| $\mathrm{err}_t$ | Miscoverage error at time $t$ | $\alpha_t^*$ | Ideal conformal parameter with $M_t(\alpha_t^*) = \alpha$ |
| $\Phi_t$ | Squared parameter tracking error | $\gamma_t$ | Dynamic adaptive step size at time $t$ |

### A.2. More Details about Conformal Prediction and Conformal Factuality

#### A.2.1. STANDARD CONFORMAL PREDICTION (SCP)

For completeness, we provide a brief review of the standard Conformal Prediction (CP) framework. Let $\{(X_i, Y_i)\}_{i=1}^n$ be a calibration dataset $\mathcal{D}_{cal}$ drawn i.i.d. from a distribution $P$. For a new test input $X_{n+1}$, CP aims to produce a set $\mathcal{C}(X_{n+1})$ such that $Y_{n+1} \in \mathcal{C}(X_{n+1})$ with probability at least $1 - \alpha$, where $\alpha \in (0, 1)$ is a user-specified error rate.

**Non-Conformity Score.** We first define a non-conformity score function $S(X, Y) \in \mathbb{R}$, which measures how "strange" a label $Y$ is given input $X$. We compute these scores for all examples in the calibration set:

$$s_i = S(X_i, Y_i), \quad \forall (X_i, Y_i) \in \mathcal{D}_{cal}. \tag{15}$$

**Calibration and Thresholding.** To achieve the $1 - \alpha$ coverage guarantee, we compute the empirical quantile of the calibration scores. Let $\hat{q}$ be the $\lceil (n + 1)(1 - \alpha) \rceil / n$-th quantile of the scores $\{s_1, \ldots, s_n\}$. The prediction set is then constructed as:

$$\hat{C}(X_{test}) = \{y \in \mathcal{Y} \mid S(X_{test}, y) \leq \hat{q}\}. \tag{16}$$

By the exchangeability assumption, this procedure guarantees marginal conformal coverage: $\mathbb{P}(Y_{test} \in \hat{C}(X_{test})) \geq 1 - \alpha$.

#### A.2.2. DETAILS OF CONFORMAL FACTUALITY (CF)

While standard CP operates on fixed output spaces, applying it to open-ended LLM generation requires bridging the gap between discrete prediction sets and natural language sequences. We adopt the Conformal Factuality (CF) framework proposed by Mohri & Hashimoto (2024), which formalizes this connection via Entailment and Back-off mechanisms.

**Entailment Function $E(\cdot)$.** The core insight of CF is to map the correctness of a generated response $Y$ to a set inclusion problem. We define the entailment set of a response $Y$ as the set of all claims implied by it:

$$E(Y) := \{y' \in \mathcal{Y} : y' \Rightarrow Y\}, \tag{17}$$

where $\Rightarrow$ denotes the logical entailment relation. Consequently, ensuring that the generated response $Y$ is factually correct with respect to the ground truth $Y^*$ is equivalent to ensuring $Y^* \in E(Y)$. This transformation allows us to apply the CP coverage guarantee $\mathbb{P}(Y^* \in \hat{C}(X)) \geq 1 - \alpha$ to language generation, where the prediction set is implicitly defined as $\hat{C}(X) = E(\hat{Y})$.

**Nested Back-off Function $F_\lambda(\cdot)$.** To construct valid prediction sets, CF employs a back-off strategy that progressively creates less specific (but more likely to be correct) outputs. Let $F_\lambda(X)$ denote a back-off function parameterized by a strictness threshold $\lambda$. In the context of LLMs, this is implemented by decomposing the initial response into a set of atomic claims $\mathcal{A} = \{a_1, \ldots, a_k\}$ and assigning a confidence score $s(a_j)$ to each. The back-off function returns the conjunction of claims that satisfy the threshold:

$$F_\lambda(X) = \mathrm{Merge}(\{a_j \in \mathcal{A} : s(a_j) \geq \lambda\}). \tag{18}$$

As $\lambda$ increases, fewer claims are retained, making the output $F_\lambda(X)$ less informative but "safer". This creates a sequence of nested entailment sets: for $\lambda_1 > \lambda_2$, we have $E(F_{\lambda_1}(X)) \supseteq E(F_{\lambda_2}(X))$.

**Calibration.** Given a calibration set $\{(X_i, Y_i^*)\}_{i=1}^n$, we compute the non-conformity score $r_i$ for each example, defined as the minimum strictness required to cover the ground truth:

$$r_i := \inf\{\lambda \in \mathbb{R} : Y_i^* \in E(F_\lambda(X_i))\}. \tag{19}$$

In practice, checking $Y_i^* \in E(F_\lambda(X_i))$ is performed by an automated factual evaluator (e.g., an LLM judge or retrieval-based system). The final calibrated threshold $\hat{\lambda}$ is chosen as the $\lceil(n+1)(1-\alpha)\rceil/n$-th quantile of the calibration scores $\{r_i\}$. During inference, the model outputs $F_{\hat{\lambda}}(X_{\text{test}})$, guaranteeing that the result entails the ground truth with probability $1 - \alpha$ (under the exchangeability assumption).

## B. Proofs of Theoretical Guarantees

In this appendix, we provide detailed proofs for the lemma and theorems in our paper.

### B.1. Proof of Lemma 4.1

**Lemma 4.1 (Restated).** *Assume the step-size policy $\gamma_t$ ensures $\gamma_t \in [\gamma_{\min}, \gamma_{\max}]$ with $\gamma_{\max} \geq \gamma_{\min} \geq 0$. Then, with probability one, the sequence $\{\alpha_t\}_{t \geq 1}$ is bounded within the interval $[-\gamma_{\max}, 1 + \gamma_{\max}]$.*

*Proof.* The proof extends the stability argument from (Gibbs & Candès, 2021) to the variable step-size setting. Recall the update rule:

$$\alpha_{t+1} = \alpha_t + \gamma_t(\alpha - \text{err}_t). \tag{20}$$

**Case 1: Lower Bound ($\alpha_t \geq -\gamma_{\max}$).** First, consider the behavior when $\alpha_t < 0$. By the definition of conformal quantiles, a negative target error rate implies the prediction set must cover the entire output space ($\hat{C}_t = \mathcal{Y}$), resulting in zero realized error ($\text{err}_t = 0$). The update rule simplifies to:

$$\alpha_{t+1} = \alpha_t + \gamma_t(\alpha - 0) = \alpha_t + \gamma_t\alpha. \tag{21}$$

Since $\alpha \in [0, 1]$ and $\gamma_t \geq 0$, we have $\alpha_{t+1} \geq \alpha_t$. This implies that once the parameter falls below 0, it strictly increases. Therefore, the parameter can only cross the lower bound $-\gamma_{\max}$ coming from a value $\alpha_{t-1} \geq 0$.

Consider the step $t$ where the parameter crosses into the negative region (i.e., $\alpha_{t-1} \geq 0$). The maximum possible decrease occurs when the error is maximal ($\text{err}_{t-1} = 1$). The update is:

$$\alpha_t = \alpha_{t-1} + \gamma_{t-1}(\alpha - 1). \tag{22}$$

Since $\alpha \geq 0$, the term $(\alpha - 1) \geq -1$. Combining this with $\alpha_{t-1} \geq 0$:

$$\alpha_t \geq 0 + \gamma_{t-1}(-1) \geq -\gamma_{\max}. \tag{23}$$

Thus, the parameter is lower-bounded by $-\gamma_{\max}$.

**Case 2: Upper Bound ($\alpha_t \leq 1 + \gamma_{\max}$).** Similarly, consider the behavior when $\alpha_t > 1$. The algorithm produces an empty set, so the realized error is maximal ($\text{err}_t = 1$). The update rule becomes:

$$\alpha_{t+1} = \alpha_t + \gamma_t(\alpha - 1). \tag{24}$$

Since $\alpha \leq 1$, the term $(\alpha - 1)$ is non-positive, implying $\alpha_{t+1} \leq \alpha_t$. The parameter strictly decreases when it exceeds 1.

Now, consider the step where the parameter crosses 1 from below (i.e., $\alpha_{t-1} \leq 1$). The maximum possible increase occurs when the error is minimal ($\text{err}_{t-1} = 0$). The update is:

$$\alpha_t = \alpha_{t-1} + \gamma_{t-1}\alpha. \tag{25}$$

Since $\alpha \leq 1$ and $\alpha_{t-1} \leq 1$:

$$\alpha_t \leq 1 + \gamma_{t-1}(1) \leq 1 + \gamma_{\max}. \tag{26}$$

Thus, the parameter is upper-bounded by $1 + \gamma_{\max}$.

Combining both cases, we conclude that $\alpha_t \in [-\gamma_{\max}, 1 + \gamma_{\max}]$ for all $t$. $\qquad\square$

## B.2. Proof of Theorem 4.2

**Theorem 4.2 (Restated).** *If the total adaptive weight $W_T = \sum_{t=1}^{T} \gamma_t \to \infty$, then the weighted average error converges to the target $\alpha$:* $\lim_{T \to \infty} \frac{\sum_{t=1}^{T} \gamma_t err_t}{W_T} = \alpha$.

*Proof.* We employ a telescoping sum argument. First, rearrange the update rule $\alpha_{t+1} = \alpha_t + \gamma_t(\alpha - err_t)$ to isolate the weighted realized error:

$$\gamma_t err_t = \gamma_t \alpha - (\alpha_{t+1} - \alpha_t). \tag{27}$$

Summing this equation from $t = 1$ to $T$:

$$\sum_{t=1}^{T} \gamma_t err_t = \sum_{t=1}^{T} \gamma_t \alpha - \sum_{t=1}^{T} (\alpha_{t+1} - \alpha_t) = \sum_{t=1}^{T} \gamma_t \alpha - (\alpha_{T+1} - \alpha_1). \tag{28}$$

Dividing both sides by the total cumulative weight $W_T$ and rearranging terms:

$$\frac{\sum_{t=1}^{T} \gamma_t err_t}{\sum_{t=1}^{T} \gamma_t} - \alpha = -\frac{\alpha_{T+1} - \alpha_1}{\sum_{t=1}^{T} \gamma_t}. \tag{29}$$

Taking the absolute value:

$$\left| \frac{\sum_{t=1}^{T} \gamma_t err_t}{\sum_{t=1}^{T} \gamma_t} - \alpha \right| = \frac{|\alpha_{T+1} - \alpha_1|}{\sum_{t=1}^{T} \gamma_t} = \frac{|\alpha_{T+1} - \alpha_1|}{W_T}. \tag{30}$$

By Lemma 4.1, the sequence $\{\alpha_t\}$ is bounded within $[-\gamma_{\max}, 1 + \gamma_{\max}]$. Therefore, the numerator is bounded by a finite constant $C$:

$$|\alpha_{T+1} - \alpha_1| \leq |\alpha_{T+1}| + |\alpha_1| \leq 2(1 + \gamma_{\max}). \tag{31}$$

Since $\sum_{t=1}^{T} \gamma_t \to \infty$ as $T \to \infty$, the right-hand side converges to 0. $\square$

## B.3. Proof of Theorem 4.5

We aim to minimize the expected parameter tracking error $\Phi_{t+1} = (\alpha_{t+1} - \alpha_{t+1}^*)^2$ given the current state $\mathcal{F}_t$. Recall the update rule $\alpha_{t+1} = \alpha_t + \gamma_t(\alpha - err_t)$. Let $Z_t = \alpha_t - \alpha_t^*$ be the current parameter error, noting that $Z_t^2 = \Phi_t$. Let $\Delta_t = \alpha_{t+1}^* - \alpha_t^*$ be the shift.

Expanding the expected next-step error:

$$\mathbb{E}_t[\Phi_{t+1}] = \mathbb{E}_t[(\alpha_t + \gamma_t(\alpha - err_t) - \alpha_{t+1}^*)^2] \tag{32}$$

$$= \mathbb{E}_t[(\alpha_t - \alpha_t^* + \gamma_t(\alpha - err_t) + \alpha_t^* - \alpha_{t+1}^*)^2] \tag{33}$$

$$= \mathbb{E}_t[(Z_t + \gamma_t(\alpha - err_t) - \Delta_t)^2] \tag{34}$$

$$= Z_t^2 + \mathbb{E}_t[\Delta_t^2] - 2Z_t \mathbb{E}_t[\Delta_t]$$
$$+ \gamma_t^2 \mathbb{E}_t[(\alpha - err_t)^2] + 2\gamma_t Z_t \mathbb{E}_t[\alpha - err_t] - 2\gamma_t \mathbb{E}_t[(\alpha - err_t)\Delta_t]. \tag{35}$$

We seek to bound the terms involving $\gamma_t$ to construct a convex upper bound for the objective:

1. **Restorative Force:** Under Assumption 4.4, the expected gradient aligns with the parameter error:

$$\mathbb{E}_t[\alpha - err_t] = \alpha - M_t(\alpha_t) = M_t(\alpha_t^*) - M_t(\alpha_t).$$

   Multiplying by $2\gamma_t Z_t$ (and noting the sign change):

$$2\gamma_t Z_t \mathbb{E}_t[\alpha - err_t] \leq -2\rho\gamma_t Z_t^2 = -2\rho\gamma_t \Phi_t.$$

2. **Variance Term (Upper Bound):** Since $err_t \in [0, 1]$ and $\alpha \in [0, 1]$, the squared error term is naturally bounded: $(\alpha - err_t)^2 \leq 1$. Thus:

$$\gamma_t^2 \mathbb{E}_t[(\alpha - err_t)^2] \leq \gamma_t^2.$$

Substituting these bounds into Eq. (35) and collecting terms independent of $\gamma_t$ into a constant $C$, we obtain a surrogate objective function $g(\gamma_t)$ that upper-bounds the expected error:

$$\mathbb{E}_t[\Phi_{t+1}] \leq C + \gamma_t^2 - \gamma_t(2\rho\Phi_t + 2\mathbb{E}_t[(\alpha - \text{err}_t)\Delta_t]) := g(\gamma_t). \tag{36}$$

Minimizing $g(\gamma_t)$ with respect to $\gamma_t$ by setting $\frac{dg}{d\gamma_t} = 0$:

$$2\gamma_t - (2\rho\Phi_t + 2\mathbb{E}_t[(\alpha - \text{err}_t)\Delta_t]) = 0 \tag{37}$$
$$\implies \gamma_t^* = \rho\Phi_t + \mathbb{E}_t[(\alpha - \text{err}_t)\Delta_t]. \tag{38}$$

Therefore, the surrogate-optimal step size scales linearly with the instantaneous parameter error $\Phi_t$ and with the latent drift $\Delta_t$ through the signed feedback coupling $\mathbb{E}_t[(\alpha - \text{err}_t)\Delta_t]$.

## C. Experiments Details

### C.1. Synthetic Data Simulation Details

In this section, we provide the specific parameters and generation processes used in the synthetic experiments described in Section 6.1.

**Data Generation.** The data stream consists of $T = 6000$ steps. The target coverage probability is set to $1 - \alpha = 0.9$. The observation $Y_t$ is drawn from $\mathcal{N}(\mu_t, 1)$.

**Shift Scenarios.** The time-varying mean parameter $\mu_t$ is generated as follows:

- **Abrupt Shift:** The mean oscillates between a base state $\mu_t = 0$ (duration 800 steps) and discrete shift states (duration 200 steps). The shift magnitudes are drawn from the set $\mu \in \{0.5, -0.5, 1.0, -2.0\}$.

- **Smooth Shift:** Following (Gibbs & Candès, 2024), the mean evolves via a momentum-based random walk:

$$\mu_t = \mu_{t-1} + \frac{1}{2}(\mu_{t-1} - \mu_{t-2}) + \frac{1}{2}\epsilon_t, \tag{39}$$

  where Gaussian noise $\epsilon_t \sim \mathcal{N}(0, 0.003)$ is injected every 5 steps to simulate gradual drift.

**Proactive Shift Signal Calculation.** For the synthetic experiment, we treat the scalar observation $X_t$ as the embedding. The background reference $\hat{\mu}_{bg,t}$ is estimated online using a sliding window of size $m = 50$:

$$\hat{\mu}_{bg,t} = \frac{1}{m}\sum_{j=1}^{m} X_{t-j}. \tag{40}$$

The shift signal is derived from the deviation of $X_t$ from the calibration center ($\mu_{in} = 0$) relative to this background.

**Evaluation Metrics Definitions.**

- **RMSE:** We calculate the Root Mean Squared Error between the local realized error rate $\hat{\pi}_t$ (estimated via a sliding window of size $W = 100$) and the target $\alpha$:

$$\text{RMSE} = \sqrt{\frac{1}{T}\sum_{t=1}^{T}(\hat{\pi}_t - \alpha)^2}. \tag{41}$$

- **Recovery Time ($T_{rec}$):** Defined as the number of steps required for the local error rate to return to the range $[\alpha - 0.05, \alpha + 0.05]$ after an abrupt distribution shift.

## C.2. Construction of Online Distribution Shifts

To rigorously evaluate PACE under non-stationary conditions, we construct synthetic data streams that simulate realistic deployment challenges. The streams operate over a horizon of $T = 6000$ steps, following an intermittent shift pattern: stable periods of 800 In-Distribution (ID) steps are punctuated by sudden 200-step bursts of Out-Of-Distribution (OOD) queries.

**MMLU** We use the dataset provided in (Ye et al., 2024) to simulate drastic semantic shifts by partitioning the dataset based on subjects. For a given experimental run, we select a specific subject (e.g., Abstract Algebra) and split its samples evenly: 50% form the calibration set $\mathcal{D}_{cal}$, and the remaining 50% serve as the ID test stream. The OOD blocks are constructed by randomly sampling questions from disjoint subjects, forcing the model to adapt to sudden changes in domain logic and difficulty.

**WikiData** Following the protocol in (Cherian et al., 2024), we simulate shifts from "head" to "tail" knowledge by stratifying entities based on their Wikipedia page view counts—a proxy for entity popularity and model familiarity. ID (Tier 1): We select the top 25% most popular entities. Half are used for calibration, and the other half constitute the stable ID stream. OOD (Tiers 2-4): The remaining entities are bucketed into three difficulty tiers: Tier 2 (25%-50%), Tier 3 (50%-75%), and Tier 4 (bottom 25%). Lower popularity typically correlates with higher hallucination rates. The stream cycles through these OOD tiers (Tier 2 → Tier 3 → Tier 4 → Tier 2 . . . ) during the shift blocks, creating a dynamic environment with varying intensities of distribution shift.

## C.3. Details

**Scoring Functions.** (1) MMLU (QA Task). Following (Lin et al., 2025; Ye et al., 2024), we extract the log-probabilities of the candidate tokens (A–F) and apply a softmax function. We adopt the Least Ambiguous Classifier (LAC) nonconformity score, defined as: $\mathcal{S}(X, Y) = 1 - f(X)_Y$, where $f(X)_Y$ denotes the softmax probability assigned to the label $Y$. (2) WikiData (Generation Task). We follow (Mohri & Hashimoto, 2024) to compute the LLM confidence and frequency scores. We sample $k = 5$ alternative output sequences at temperature $T = 1.0$ and utilize `GPT-5.2` to count the recurrence frequency of each atomic claim across these samples. The score reflects the consistency of the generated claim.

**Ground Truth Acquisition.** To acquire the ground truth $Y_t$ for open-ended generation, we employ an automated oracle mimicking human fact-checking. Following the FActScore procedure (Min et al., 2023), the generated response $\hat{Y}_t$ is first parsed into atomic claims by `GPT-4o-mini`. Each claim is then verified against retrieved Wikipedia passages using a frontier LLM (`GPT-5.2`) as the judge.

**Models.** For MMLU, we use `QWEN-72B` as the backbone. For WikiData, biographies are generated using `GPT-5.2`. The decomposition and verification steps utilize `GPT-4o-mini` and `GPT-5.2` respectively to ensure high-quality evaluation.

## C.4. Sensitivity Analysis

### C.4.1. Synthetic Data Simulation

To rigorously evaluate the synergistic contribution of the proactive and reactive components, we conducted a controlled ablation study by parameterizing the mixing weights as a convex combination $c_1 = 1 - c_2$ and varying the reactive weight $c_2$ across the interval $[0, 1]$. This setup continuously interpolates between a purely reactive policy ($c_2 = 0$) and a purely proactive one ($c_2 = 1$). As illustrated in Figure 4, this "U-shaped" trajectory empirically validates the necessity of dual-signal fusion: while $S_{\text{shift}}$ enables rapid adaptation to abrupt distribution jumps, $S_{\text{error}}$ is essential for precise calibration during stationary periods; combining them yields a lower tracking error than relying on either signal in isolation. Crucially, PACE consistently outperforms the baseline methods which are depicted as horizontal reference lines—across the entire hyperparameter spectrum. Even in the limiting single-signal regimes, our framework maintains a significant performance margin over the strongest reactive baselines DtACI.

### C.4.2. Real-World Datasets

To understand the contribution of the proactive signal $S_{shift}$ versus the reactive signal $S_{error}$, we conducted a sensitivity analysis by parametrizing the update rule as a convex combination: $\gamma_t = (1 - w) \cdot S_{error} + w \cdot S_{shift}$. We varied $w \in [0, 1]$ and analyzed the impact on both tracking fidelity (RMSE) and downstream utility (Claim Retention).

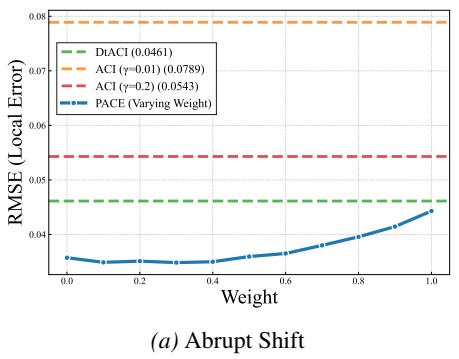

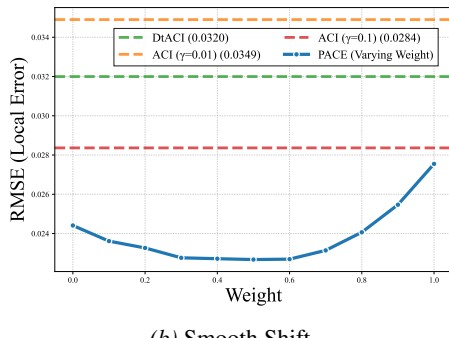

*(a)* Abrupt Shift

*(b)* Smooth Shift

*Figure 3.* Parameter ablation analysis. We vary the weight of $c_2$ (x-axis) versus local error rate RMSE. The "U-shaped" curves demonstrate the necessity of fusing both signals: relying solely on shift detection or error tracking is suboptimal. PACE (blue solid line) consistently outperforms all baselines (horizontal dashed lines) across a wide range of hyperparameter configurations.

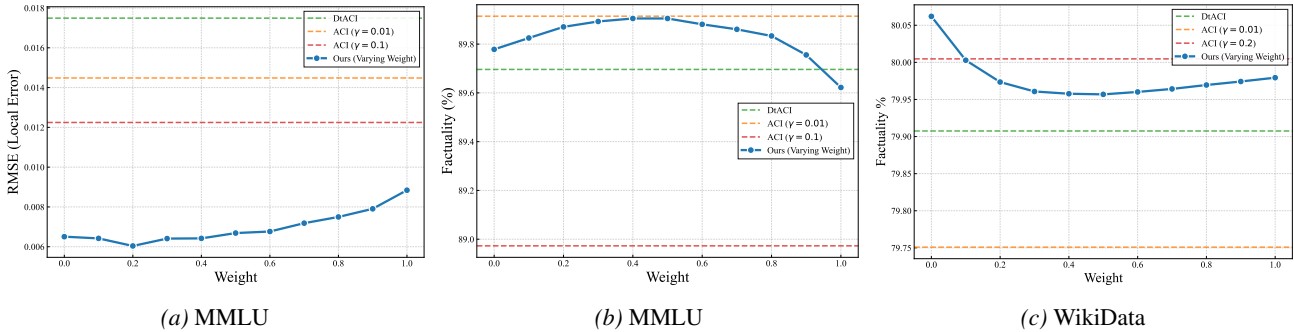

*(a)* MMLU                 *(b)* MMLU                 *(c)* WikiData

*Figure 4.* Parameter sensitivity analysis on real-world datasets.

**MMLU.** As shown in Figure 4b, the MMLU dataset exhibits a classic bias-variance trade-off. Purely reactive updates ($w = 0$) suffer from lag (bias), while purely proactive updates ($w = 1$) may over-react (variance). The optimal performance is achieved in the middle region ($w \approx 0.2$), forming a "U-shaped" curve. This aligns with our synthetic simulation results, confirming that for structured QA tasks with distinct topic shifts, fusing both signals is essential for minimizing tracking error.

**WikiData.** In the WikiData experiment (Figure 4c(b) and (c)), we observe a different phenomenon: the RMSE does not follow a U-shape but increases monotonically, often exceeding static baselines like SCP or ACI($\gamma = 0.01$). This counter-intuitive result highlights the limitations of RMSE as a metric in open-ended generation tasks characterized by high *aleatoric uncertainty*. In generation tasks, the instantaneous error rate is highly stochastic. Static methods (like SCP or conservative ACI) effectively act as a "low-pass filter," predicting the long-term average error rate. In signal processing, predicting the mean is known to minimize Mean Squared Error (MSE) when the signal-to-noise ratio is low. Therefore, baselines that "do nothing" (ignore shifts) artificially achieve lower RMSE by suppressing variance.

Driven by the proactive signal, PACE actively adjusts $\alpha_t$ based on input semantics. While this active adjustment increases the variance of the parameter trajectory (leading to higher RMSE against the noisy ground truth), it provides tangible benefits in utility. As shown in Figure 4c(c), increasing the weight of the proactive signal significantly boosts the Claim Retention Rate (from $\sim 71.4\%$ to $\sim 72.3\%$). This analysis reveals that in high-noise regimes, a lower RMSE does not necessarily imply better system performance. While static baselines minimize RMSE by being conservative, they fail to relax constraints when possible. PACE's proactive mechanism, despite incurring a higher tracking variance, successfully identifies "safe" instances to widen prediction sets, thereby maximizing information retention without violating the global safety guarantee.

**Effect of sliding window size $n$.** We further study the effect of the sliding window size $n$ used to estimate the recent empirical error rate. The window size controls a standard stability-adaptivity trade-off: smaller windows react quickly but are more sensitive to noise, whereas larger windows smooth the estimate but can introduce adaptation lag after distribution shifts. We evaluate $n \in \{5, 10, 25, 50, 100\}$ on synthetic streams.

*Table 6.* Ablation study for the sliding window size $n$ under abrupt and smooth shift scenarios. Recovery is reported only for the abrupt setting.

| Window Size | Abrupt | | | Smooth | |
| --- | --- | --- | --- | --- | --- |
| | Coverage | RMSE | Recovery | Coverage | RMSE |
| $n = 5$ | 0.8931 | 0.0467 | **38.20** | 0.8963 | 0.0279 |
| $n = 10$ | 0.8911 | 0.0422 | 55.91 | 0.8929 | 0.0252 |
| $n = 25$ | 0.8943 | **0.0401** | 50.22 | 0.8947 | **0.0238** |
| $n = 50$ | 0.8966 | 0.0433 | 62.32 | 0.8974 | 0.0244 |
| $n = 100$ | **0.8972** | 0.0496 | 80.27 | **0.8986** | 0.0265 |

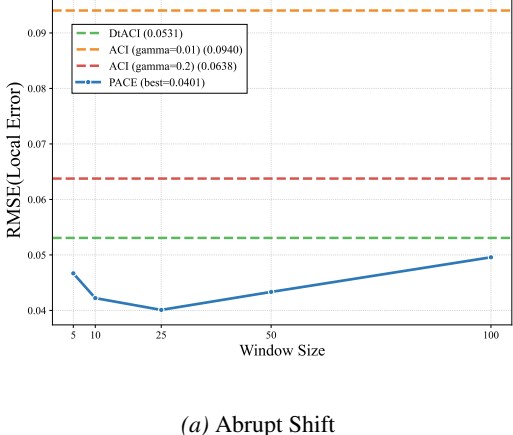

*(a)* Abrupt Shift

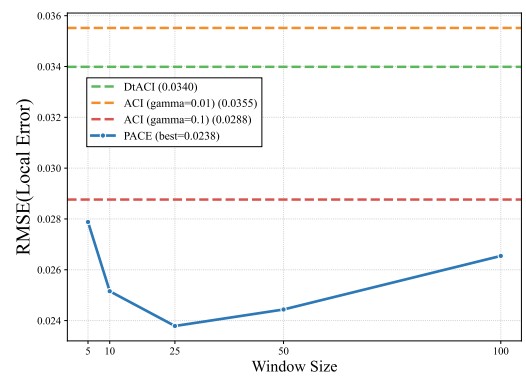

*(b)* Smooth Shift

*Figure 5.* Effect of the sliding window size $n$ on local error-rate tracking. Small windows react quickly but overreact to noise, while large windows smooth the error estimate but introduce adaptation lag. The intermediate setting $n = 25$ provides the best overall tracking balance.

Table 6 reports the quantitative results. Very small windows such as $n = 5$ recover quickly but yield higher tracking error. Large windows such as $n = 100$ produce smoother coverage estimates but suffer from larger RMSE and delayed recovery. The intermediate value $n = 25$ achieves the lowest RMSE in both settings and provides the best overall balance between stability and responsiveness.

### C.5. Sensitivity Analysis of Fixed-Step ACI on WikiData

To validate the necessity of the adaptive mechanism in PACE, we performed a fine-grained sensitivity analysis on the standard ACI baseline. We varied the fixed step size $\gamma$ from $0.01$ to $1.0$ on the WikiData stream and recorded the performance metrics. The results are summarized in Figure 6 and detailed below.

The experimental results reveal that ACI is structurally sensitive to the choice of $\gamma$, facing a rigid trade-off that cannot be resolved by static tuning. Small step sizes achieve the lowest global RMSE by smoothing out noise. However, they fail to react to shifts, resulting in unacceptable recovery times ($45 \sim 90$ steps). Large step sizes reduce recovery time ($< 25$ steps) but significantly degrade stability (RMSE rises to $0.174$) and claim retention (dropping to $\sim 70.4\%$). Our method PACE bypasses this sensitivity issue. By dynamically modulating the update magnitude based on $S_{error}$ and $S_{shift}$, PACE achieves a Claim Retention rate of $0.7227$, which strictly outperforms the "Oracle" fixed ACI ($\gamma = 0.2$). By aggressively increasing the step size only during genuine shifts and dampening it during stationary periods, PACE breaks the rigid stability-reactivity trade-off inherent to fixed-parameter ACI. In essence, PACE automatically selects the "better $\gamma$" at each time step, reducing the need to select a single fixed ACI step size that must work across both stationary and shifted regimes. The remaining weights $c_1$ and $c_2$ control the relative emphasis of two theoretically motivated proxies, and our sensitivity analysis shows stable performance across a broad range of configurations.

### C.6. Long-term Error Rate Convergence.

To empirically validate the theoretical insights discussed in **Remark 4.3 (Accelerated Adaptation)**, we evaluate the long-term performance of our proposed method on the **Wikidata** dataset. We monitor the cumulative average error rate up

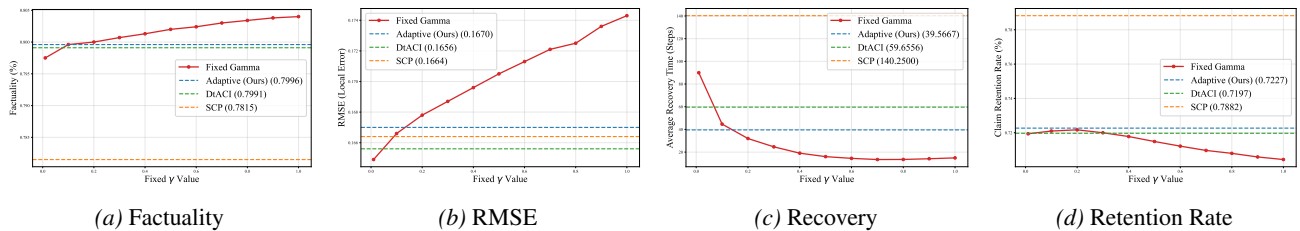

| *(a)* Factuality | *(b)* RMSE | *(c)* Recovery | *(d)* Retention Rate |

*Figure 6.* Parameter sensitivity analysis on WikiData datasets.

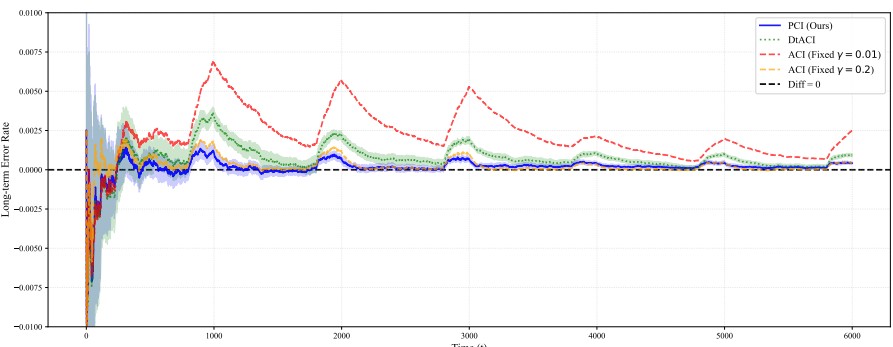

*Figure 7.* Long-term error rate convergence.

to time $t$, defined as:

$$\bar{\text{err}}_t = \frac{1}{t} \sum_{i=1}^{t} \text{err}_i, \tag{42}$$

where $\text{err}_i$ represents the factuality error (or miscoverage gap) at step $i$. This metric reflects the algorithm's ability to maintain valid coverage or factuality over an extended horizon.

As illustrated in Figure 7, the curve for PACE (Ours) (blue line) converges to the optimal error rate (indicated by the dashed black line at 0) significantly faster than the baselines. This confirms that our adaptive mechanism effectively mitigates the lag typically observed in online calibration. While ACI (Fixed $\gamma = 0.01$) (red dashed line) exhibits large oscillations and slow recovery from distribution shifts—characteristic of a learning rate that is too small for dynamic environments—our method maintains stability with much narrower variance (shaded area) while retaining the agility to correct errors promptly. Although DtACI (green dotted line) also achieves convergence, PACE demonstrates a tighter error bound and reduced volatility throughout the process.

These observations align with Remark 4.3, demonstrating that our approach does not merely achieve asymptotic validity but does so with superior sample efficiency in the finite-sample regime.

### C.7. Computational Efficiency and Online Latency Analysis

To assess the feasibility of deploying PACE in real-time, high-stakes environments, we conducted a rigorous end-to-end latency profiling. We dissect the total processing time for a single time step $t$, covering the entire lifecycle from receiving the input query $X_t$ to outputting the calibrated response $\hat{C}_t$ and updating the parameter $\alpha_{t+1}$ (as defined in Algorithm 1).

**Latency Decomposition.** The total latency $T_{total}$ can be decomposed into the shared pipeline latency ($T_{Base}$) and the method-specific algorithmic overhead ($T_{Alg}$):

$$T_{total} = \underbrace{T_{Gen} + T_{Decomp} + T_{Score}}_{T_{Base} \text{ (Shared)}} + \underbrace{T_{Adapt}}_{T_{Alg} \text{ (Method-Specific)}} \tag{43}$$

- **Shared Base Pipeline** ($T_{Base}$): This component includes the computationally intensive operations required by the Conformal Factuality framework. It involves: (1) **Generation** ($T_{Gen}$): Generating the initial biography using GPT-5.2; (2) **Sampling & Parsing** ($T_{Decomp}$): Generating 5 parallel samples for consistency checking and decomposing the text

into atomic claims using GPT-4o-mini; (3) **Scoring & Enrichment** ($T_{Score}$)**:** Retrieving evidence and performing self-evaluation and entailment checks for each claim using GPT-5.2.

- **Algorithmic Overhead** ($T_{Alg}$)**:** This component represents the marginal cost introduced by the adaptive calibration strategy. For PACE, this includes computing the Mahalanobis distance (RMD), calculating the dynamic step size $\gamma_t$, and updating $\alpha_t$.

**Results.** We measured the wall-clock time averaged over 30 independent trials with the same stream in Appendix C.2, with stream length of $T = 6000$. The breakdown is presented in Table 7.

The results show that the end-to-end latency is overwhelmingly dominated by the LLM inference and scoring process, which averages 16.47 seconds per query. In stark contrast, the algorithmic overhead of PACE is measured at 0.072 ms.

*Table 7.* **Latency Breakdown per Query.** The proposed PACE algorithm introduces negligible overhead compared to the heavy computation required for LLM generation and factuality scoring.

| Component | Operation (Algorithm 1) | Time | % of Total |
|---|---|---|---|
| **Base Pipeline** ($T_{Base}$) | **LLM Gen., Scoring, & Decomp.** | **16.466 s** | **99.999%** |
| $\hookrightarrow$ *Bio Generation* | *GPT-5.2 (Full Generation)* | *4.34 s* | – |
| $\hookrightarrow$ *Parallel Block* | *Sampling ($\times 5$) & Parsing* | *5.04 s* | – |
| $\hookrightarrow$ *Scoring* | *Freq. Score & Enrichment* | *7.08 s* | – |
| **Algorithmic Overhead** ($T_{Alg}$) | **Update $\gamma_t$ & $\alpha_t$** | | |
| $\hookrightarrow$ Standard ACI | *Fixed Step Update* | 0.0604 ms | $< 10^{-5}\%$ |
| $\hookrightarrow$ DtACI | *Gradient-based Update* | 0.0731 ms | $< 10^{-5}\%$ |
| $\hookrightarrow$ **PACE (Ours)** | *RMD + Dynamic Update* | **0.0724 ms** | $< 10^{-5}\%$ |

**Discussion.** Despite the complexity of calculating the semantic shift signal (RMD) and the dual-signal update rule, PACE remains extremely efficient. The computation of RMD operates on low-dimensional embedding statistics and utilizes efficient vector operations, resulting in an overhead virtually identical to the simpler DtACI baseline and only marginally higher than the static ACI. Crucially, the 0.0724 ms overhead is five orders of magnitude smaller than the base pipeline latency. This indicates that PACE provides efficient robustness: it achieves a ~40% reduction in tracking error and rapid adaptation to distribution shifts (as shown in Sec. 7) with zero perceptible impact on the system's overall response time. This confirms that PACE is well-suited for online deployment where the bottleneck lies in model inference, not calibration logic.

### C.8. Additional Robustness Experiments

#### C.8.1. SAOCP COMPARISON

We further compare PACE with SAOCP (Bhatnagar et al., 2023), a strongly adaptive online conformal prediction method based on interval experts. SAOCP is a competitive reactive baseline because it can rapidly discard stale pre-shift history in piecewise-stationary environments. Tables 8a and 8b report results on the abrupt and smooth synthetic streams, respectively, and Fig. 8 visualizes the corresponding local error-rate trajectories. Under abrupt shifts, SAOCP achieves fast recovery but suffers from substantially larger worst-case violations at shift onsets. Under smooth shifts, where there are no clear stationary segments for interval experts to exploit, PACE achieves tighter tracking and smaller worst-case violation. These results suggest that PACE's proactive shift signal complements reactive feedback, especially when controlling transient violations is important.

*Table 8.* Comparison with SAOCP on synthetic data streams. *Worst Violation* denotes the maximum deviation of the local error rate from the target.

*(a)* Results under **Abrupt** shifts.

| Method | Coverage | RMSE | Recovery | Worst Violation |
|---|---|---|---|---|
| **DtACI** | 0.8862 | 0.0461 | 120.66 | 0.2754 |
| **SAOCP** | 0.8857 | **0.0293** | **21.11** | 0.4556 |
| **PACE** | **0.8953** | 0.0345 | 54.18 | **0.2507** |

*(b)* Results under **Smooth** shifts.

| Method | Coverage | RMSE | Worst Violation |
|---|---|---|---|
| **DtACI** | 0.8948 | 0.0340 | 0.2305 |
| **SAOCP** | 0.8873 | 0.0267 | 0.4556 |
| **PACE** | **0.8982** | **0.0233** | **0.2232** |

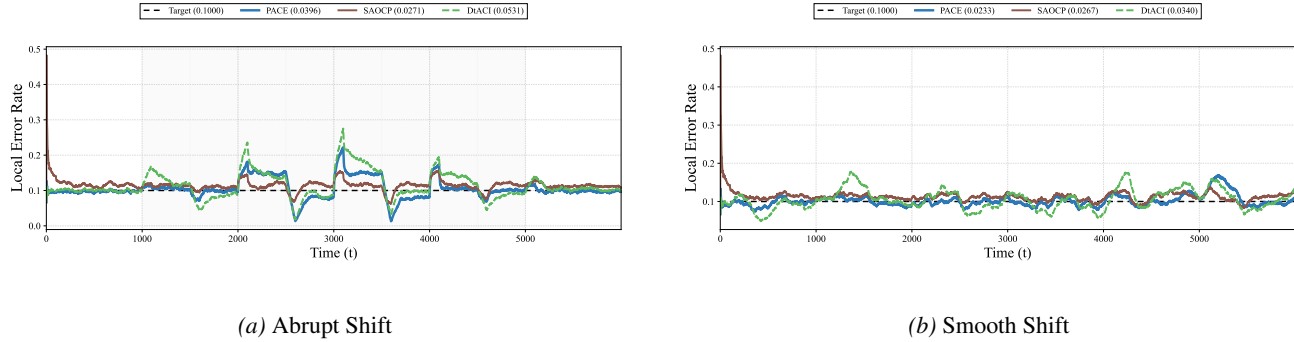

*(a)* Abrupt Shift                                                        *(b)* Smooth Shift

*Figure 8.* Comparison with SAOCP on synthetic data streams.

### C.8.2. ROBUSTNESS UNDER COMPLEX MIXTURE SHIFTS

We evaluate PACE under a more complex synthetic stream with a multi-component mixture distribution. The stream contains three Gaussian components with increasing difficulty, each with distinct prompt embeddings:

- **Easy**: $Y \sim \mathcal{N}(-0.3, 1)$, so $\alpha_t^* > 0.1$;
- **Medium**: $Y \sim \mathcal{N}(0, 1)$, so $\alpha_t^* \approx 0.1$;
- **Hard**: $Y \sim \mathcal{N}(0.7, 1)$, so $\alpha_t^* < 0.1$.

At each time step, a sample is drawn from the mixture $w_t^{\text{easy}} \cdot \text{Easy} + w_t^{\text{med}} \cdot \text{Medium} + w_t^{\text{hard}} \cdot \text{Hard}$, where the weights undergo abrupt regime transitions every 1000 steps: $(0.1, 0.8, 0.1) \to (0.1, 0.1, 0.8) \to (0.8, 0.1, 0.1) \to \cdots$. This creates shifts where $\alpha_t^*$ both increases and decreases, simulating realistic deployment settings where query difficulty fluctuates over time.

Table 9 reports the quantitative results, and Fig. 9a shows the local error-rate trajectories. PACE achieves the lowest RMSE, fastest recovery, and smallest worst violation, while maintaining coverage close to the target level. Note that ACI ($\gamma = 0.01$) has the coverage closest to the target in this setting; PACE does not optimize for marginal coverage alone but for joint tracking fidelity and violation control. SAOCP recovers quickly, but it incurs substantially larger transient violations, consistent with its reactive interval-expert mechanism. These results suggest that PACE remains effective under nonlinear mixture dynamics and regime-switching difficulty patterns.

*Table 9.* Robustness under a multi-component mixture distribution with regime-switching difficulty levels. *Worst Violation* denotes the maximum deviation of the local error rate from the target error rate.

| Method | Coverage | RMSE | Recovery | Worst Violation |
|---|---|---|---|---|
| **SCP** | $0.8539 \pm 0.0049$ | $0.0967 \pm 0.0054$ | 272.79 | $0.3120 \pm 0.1628$ |
| **ACI ($\gamma = 0.01$)** | $\mathbf{0.9002 \pm 0.0005}$ | $0.0266 \pm 0.0020$ | 25.11 | $0.2009 \pm 0.2077$ |
| **ACI ($\gamma = 0.2$)** | $0.8648 \pm 0.0018$ | $0.0460 \pm 0.0021$ | 92.49 | $0.1997 \pm 0.2006$ |
| **DtACI** | $0.8972 \pm 0.0014$ | $0.0302 \pm 0.0016$ | 26.05 | $0.2079 \pm 0.2043$ |
| **SAOCP** | $0.8879 \pm 0.0019$ | $0.0222 \pm 0.0018$ | 2.51 | $0.4389 \pm 0.1268$ |
| **PACE (Ours)** | $0.8940 \pm 0.0013$ | $\mathbf{0.0191 \pm 0.0013}$ | **1.23** | $\mathbf{0.1827 \pm 0.2159}$ |

### C.8.3. ROBUSTNESS UNDER TIME-VARYING BETA SHIFTS

We further evaluate PACE under a non-Gaussian bounded synthetic stream based on time-varying Beta distributions. The calibration distribution is $\text{Beta}(2, 2)$, and we define three regimes with symmetric difficulty offsets: Easy with $Y \sim \text{Beta}(2, 3)$, Medium with $Y \sim \text{Beta}(2, 2)$, and Hard with $Y \sim \text{Beta}(3, 2)$. Each regime is associated with a distinct prompt embedding center, allowing the proactive shift score to detect changes in the input distribution. The stream alternates among the three regimes every 500 steps over a horizon of $T = 6000$.

Table 10 reports the quantitative results, and Fig. 9b shows the local error-rate trajectories. PACE achieves the lowest RMSE and the smallest worst violation while maintaining coverage close to the target level. Although ACI with a large fixed step size recovers faster immediately after transitions, it exhibits less stable tracking and larger worst-case violations than PACE. These results show that PACE remains effective beyond Gaussian synthetic settings and provides stable local error control under bounded, time-varying Beta shifts.

*Table 10.* Robustness under time-varying Beta distributions.

| Method | RMSE | Recovery | Coverage | Worst Violation |
|---|---|---|---|---|
| **SCP** | 0.0751 | 85.05 | 0.9000 | 0.2332 |
| **ACI** ($\gamma = 0.01$) | 0.0450 | 65.78 | **0.9001** | 0.2360 |
| **ACI** ($\gamma = 0.2$) | 0.0313 | **4.68** | 0.8787 | 0.1941 |
| **DtACI** | 0.0376 | 38.13 | 0.8970 | 0.2027 |
| **PACE (Ours)** | **0.0300** | 19.93 | 0.8979 | **0.1889** |

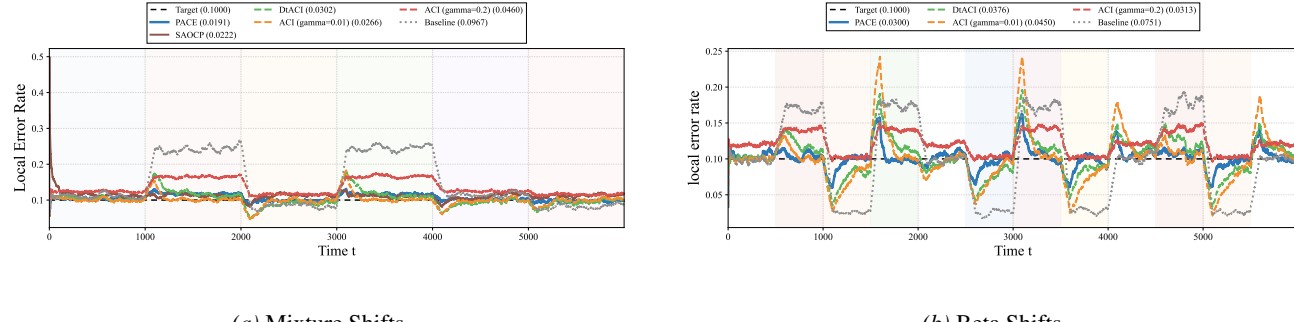

*(a)* Mixture Shifts                    *(b)* Beta Shifts

*Figure 9.* Local error-rate tracking under additional robustness settings. (a) Multi-component mixture distribution with regime switching: PACE maintains tight tracking around the target across regime transitions, while reactive baselines exhibit larger spikes at transition points. (b) Time-varying Beta distributions alternating among Easy, Medium, and Hard regimes every 500 steps: PACE tracks the target closely across both upward and downward shifts, while fixed-step ACI variants exhibit overshoot or persistent over-coverage.

## C.9. Prediction Set Size Dynamics

We further analyze how the prediction set size evolves over the MMLU stream to verify that PACE adapts not only its error rate but also its output granularity in response to distribution shifts. A well-calibrated adaptive method should automatically expand prediction sets when the underlying difficulty increases (e.g., during out-of-distribution topic blocks) and contract them once the stream returns to familiar territory. This dynamic is desirable because it reflects genuine uncertainty rather than a fixed level of conservatism.

Fig. 10 shows the rolling-window prediction set size for each method over the MMLU stream. PACE expands prediction sets during OOD blocks and contracts them during easier in-distribution phases, tracking the stream difficulty more faithfully than fixed-step ACI variants. In contrast, ACI with a large step size $\gamma$ tends to over-expand during easy phases, while ACI with a small $\gamma$ responds too slowly during hard phases. SCP maintains a constant set size throughout, reflecting its inability to adapt to distributional changes. These results confirm that PACE's adaptive step size translates into appropriately sized prediction sets across varying difficulty regimes.

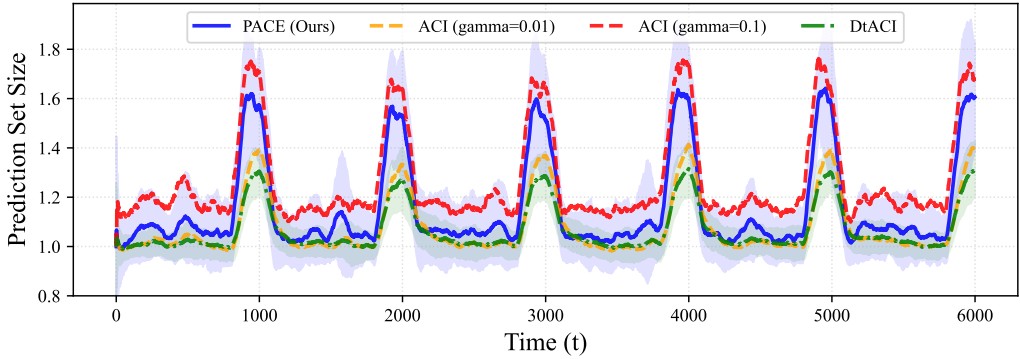

*Figure 10.* Prediction set size dynamics on the MMLU stream.

## C.10. Additional Base Model Results on MMLU

To evaluate whether the observed gains of PACE are specific to a single base model, we repeat the MMLU experiment using Llama-2-70B as the underlying language model. The nonconformity score and evaluation protocol are identical to the main experiments; only the base LLM changes. Table 11 reports factuality, tracking RMSE, recovery steps, and average prediction set size for all methods.

PACE again achieves the lowest tracking RMSE (0.0037), a reduction of 36–55% relative to the best reactive baseline, while maintaining factuality close to the 90% target. Recovery steps are competitive with the fast-adapting ACI ($\gamma = 0.1$) variant, and the average set size remains modest. These results indicate that PACE's advantages are consistent across base models and are not an artifact of a particular LLM's output distribution.

*Table 11.* MMLU results using Llama-2-70B as the base language model. Target error rate $\alpha = 0.1$. **Bold** denotes the best value in each column.

| Method | Factuality (%) | RMSE | Recovery | Set Size |
|---|---|---|---|---|
| **SCP** | 90.36 | 0.0081 | 283.79 | 1.58 |
| **ACI** ($\gamma = 0.01$) | **89.99** | 0.0058 | 85.66 | **1.55** |
| **ACI** ($\gamma = 0.1$) | 89.32 | 0.0072 | **22.14** | 1.73 |
| **DtACI** | 90.05 | 0.0058 | 76.88 | **1.55** |
| **PACE (Ours)** | 89.94 | **0.0037** | 28.01 | 1.64 |

