# OpenReview forum: "Beyond Reactivity: Proactive Adaptive Conformal Inference for Online LLM Factuality"
_ICML.cc/2026/Conference — ICML 2026 regular_

### Official Review · Reviewer_A1qW · 2026-03-12

**Soundness:** 3
**Presentation:** 3
**Significance:** 3
**Originality:** 3
**Overall Recommendation:** 4
**Confidence:** 4

**Summary:**

This paper proposes a novel conformal inference framework for LLM facing distribution shift. This method has faster adaptation before errors accumulate with theoretical guarantees. Empirically, sufficient evaluations verify the effectiveness of the proposed method.

**Compliance With Llm Reviewing Policy:**

Affirmed.

**Key Questions For Authors:**

1. It is not entirely clear how conformal prediction is implemented for sequence generation in this work. In particular, I would like clarification on whether the method performs token-level verification or sequence-level verification.
2. Can this distribution assumptions (assumption 4.4) be further relaxed? Actually, this assumption probably holds in practice.

**Limitations:**

it would be better if the authors discussed the limitations of the proposed method.

**Strengths And Weaknesses:**

Strengths

1. This method is well-motivated. Traditional  conformal prediction is not applicable to  distribution shift situation. This method tackles this challenge.

2. The theoretical guarantees are good, although I haven’t checked all the proof details.

3. This paper is well-written and easy to follow.

Weakness

1. The assumptions are strong, especially the assumption on distribution shift. It only captures some special scenarios. Also, the Assumption 4.4 requires the distribution drift aligns with the gradient update direction in expectation. It is hard to satisfy and understand.

2. There is still gap between theoretical findings and PACE algorithm (Eq. 14). The global parameters $c_1$ and $c_2$ are not inspired by the theoretical results.

3. The empirical evaluations are insufficient. More tasks and dataset are required.

---

> ### Author Rebuttal · Authors · 2026-03-31
>
> We thank Reviewer A1qW for the careful assessment and valuable questions.
> > Weakness 1: Strong assumptions, especially Assumption 4.4
>
> The alignment condition states that $$E\_t[(\alpha - \text{err}\_t)\Delta\_t] \geq \kappa\sigma\_{\Delta,t}.$$ This condition captures a natural property of conformal prediction. Consider two shift directions: (1) errors increase ($\alpha - \text{err}_t < 0$) while the optimal threshold must tighten ($\Delta_t < 0$), giving a positive product. (2) errors decrease ($\alpha - \text{err}_t > 0$) while the threshold can relax ($\Delta_t > 0$), again positive. The primary violation scenario is an instantaneous reversal the stream is currently hard but becomes easy at the very next step which is transient and mitigated by the expectation over $\mathcal{F}\_t$.
>
> Motivated by the reviewer's question, we have derived a more general Theorem 4.5 that does not require Assumption 4.4(b). The key insight is to directly minimize the exact quadratic form of $E_t[\Phi_{t+1}]$ (Eq. 35 in Appendix B.3) rather than constructing a surrogate upper bound. Under only Assumption 4.4(a), the optimal step size becomes:
> $$\gamma_t^* \propto \rho\Phi_t + E_t[(\alpha - \text{err}_t)\Delta_t]$$
> The second term is naturally signed: positive when error feedback aligns with the drift (the common case above), negative when anti-aligned. Both are optimal and require no directional assumption.
>
> The original Thm 4.5 is recovered as a special case when alignment holds. PACE's algorithm (Eq. 14) and all experiments are completely unaffected. The generalized theorem shows that PACE's design estimating magnitude via $S\_{\text{shift}}$
>  and direction via $(\alpha - \text{err}\_t)$ is principled under arbitrary drift.
>
> > Weakness 2: Gap between theory and PACE algorithm
>
> We'd like to clarify that $c_1$ and $c_2$ are directly motivated by the theory, and the gap is narrower than it may appear.
>
> The optimal step size takes the form $\gamma_t^* = \rho \Phi_{t}+ \kappa \sigma_{\Delta, t}$. PACE instantiates this as $\gamma_t = c_1 S_{\text{error}} + c_2 S_{\text{shift}}$. The global coefficients $c_1$ and $c_2$ approximate the environment-dependent constants $\rho$ (the strong-monotonicity parameter of $M_t(\cdot)$) and $\kappa$ (the alignment coefficient). These constants depend on the unknown miscoverage function $M\_(t)$ and are not directly computable.
>
> Importantly, our ablation studies (Fig. 3-4, Appendix C.4) demonstrate that PACE consistently outperforms all baselines across a wide range of $(c_1,c_2)$ configurations.
>
> > Weakness 3: Insufficient empirical evaluation
>
> We have substantially expanded the evaluation in response to all reviewers' feedback:
> 1. We added SAOCP (Improved Online Conformal Prediction via Strongly Adaptive Online Learning, ICML 2023), a competitive method that outperforms DtACI in its original paper. On synthetic data, PACE achieves lower worst-case violation (0.25 vs. 0.46) under abrupt shifts and strictly outperforms SAOCP across all metrics under smooth shifts (see our response to Reviewer VfHU, Weakness 2).
> 2. We added Llama-2-70B on MMLU. PACE achieves the lowest RMSE (0.0037) (see our response to Reviewer Feh9, Key Question 2).
> 3. (1) A mixture distribution where PACE also achieves the lowest RMSE (see our response to Reviewer qV1y). (2) A time-varying adversarial Beta distribution, where PACE achieves the lowest RMSE and worst-case violation (see our response to Reviewer Feh9, Weakness 2).
> In total, the revised evaluation covers 3 LLMs (Qwen-72B, Llama-2-70B, GPT-5.2), 2 real-world tasks (MMLU, WikiData), 4 synthetic settings (abrupt, smooth, Beta, mixture), and 5 baselines (SCP, ACI, DtACI, SAOCP, CF).
>
> > Key Question 1: Token-level vs sequence-level verification
>
> Our method operates at the option-level or claim-level, not token-level:
> - Multi-Choice QA (MMLU): It performs sequence-level (option-level) verification. The prediction set directly consists of a subset of the candidate labels (e.g., options A-F) based on the model's sequence-level confidence.
> - Open-Ended Generation (WikiData): It performs sequence-level (claim-level) verification following the Conformal Factuality framework. The LLM first generates a complete raw response, which is then decomposed into atomic claims. Conformal filtering is applied post-generation to retain only the factually valid claims. We will detail these implementations in Sections 3.2 and 5.2.
>
> > Missing limitations
> 1. PACE currently assumes that atomic claims within a single response are evaluated independently, which is violated in multi-step reasoning where errors propagate across steps.
> 2. PACE is designed for knowledge-intensive text tasks. Extending to multimodal LLMs where factuality spans visual grounding and cross-modal consistency introduces new dimensions beyond the current framework.
> 3. Like all ACI-family methods, PACE provides asymptotic coverage convergence. Non-asymptotic bounds under arbitrary shifts remain an open challenge.

---

> > ### Author Rebuttal · Reviewer_A1qW · 2026-04-04
> >
> > Thanks for your response. I decide to keep my original score.

---

> > > ### Author Response · Authors · 2026-04-05
> > >
> > > Thank you again for your time and the constructive feedback you provided. We deeply appreciate your effort in reviewing our paper and rebuttal.

---

### Official Review · Reviewer_Feh9 · 2026-03-12

**Soundness:** 3
**Presentation:** 3
**Significance:** 3
**Originality:** 2
**Overall Recommendation:** 4
**Confidence:** 3

**Summary:**

The paper investigates conformal factuality control for online LLM deployment setting under distribution shift. Specifically, the paper proposes PACE, which aims to adapt the conformal parameter online with a dynamic step size, instead of using a fixed update rate like standard ACI methods. The paper theoretically justifies the optimal step size should scale linearly with the
current parameter error and the shift magnitude. Based on this, the paper proposes to use two signals to update the $\alpha_t$: 1) the deviation between observed error and target error rate $S_{error}$ 2) the distribution shift magnitude, which is measured by prompt embeddings and relative mahalonobis distance $S_{shift}$. Empirically, the paper experimented on both synthetic data simulation with two types of distribution shifts and real-world LLM experiments with Qwen-72B for MMLU and GPT-5.2 for WikiData. On synthetic data, PACE achieves best recovery than baselines on the abrupt shift settings. On MMLU, PACE achieves the best RMSE and on WikiData, PACE achieves comparable results.

**Compliance With Llm Reviewing Policy:**

Affirmed.

**Final Justification:**

Overall I think this is a well-motivated and technical solid paper. During the rebuttal, I think the authors take my suggestions seriously and address my concerns. I maintain my original score for recommending acceptance.

**Key Questions For Authors:**

1. How sensitive PACE is to the configurations of $c_1$ and $c_2$? How to tune these two hyperparameters?
2. It will be good to include more models for the evluations.

**Limitations:**

The paper does not provide a sufficiently explicit or thorough discussion of its limitations. It includes statement of social impact.

**Strengths And Weaknesses:**

**Strength**
1. The paper is well motivated: calibrating LLM factuality in online deployment under query distribution shift is an important and realistic problem.
2. The method is technically reasonable and lightweight. It only adapts the step size using two signals, which makes it simple and potentially practical.
3. The experiments are comprehensive by considering both synthetic and real-world settings on two LLMs. And the results on the synthetic data shows strong performance.
4. The real-world shift settings are also reasonably diverse, including both domain shift and entity/popularity shift.

**Weakness**
1. The proactive signal is based on Relative Mahalanobis Distance in prompt embedding space, which seems to capture semantic novelty more than factuality difficulty. These are not the same: a prompt may be far from the calibration set yet still be easy and factual, while a prompt that looks semantically similar may be much more hallucination-prone. It may therefore be more appropriate to incorporate uncertainty-related features or other signals more directly tied to factuality risk. Along the same line, Assumption 4.4 should be better justified or empirically examined, especially regarding whether the proposed shift proxy is actually aligned with the update direction in practice.
2. In the simulation study, it would be helpful to include a more adversarial setting. Real users can phrase queries in ways that encourage speculation or unsupported generation, which is not well captured by a smooth momentum-based random walk. For example, you can consider using a time-varying Beta distribution $\beta(a_t,b)$ with more severe distribution shift over time.

---

> ### Author Rebuttal · Authors · 2026-03-31
>
> We thank the reviewer for the technically precise and insightful review.
> > Weakness 1: RMD captures semantic novelty, not factuality difficulty
>
> Semantic novelty and factuality difficulty are distinct, and RMD is designed to capture only the former. In PACE's update rule $\alpha_{t+1} = \alpha_t + \gamma_t(\alpha - \text{err}_t)$, the two signals serve distinct roles. RMD estimates the shift magnitude $|\Delta_t|$ only, i.e., how far the current stream has drifted from the calibration distribution. Factuality correctness is automatically handled by the reactive error term $(\alpha - \text{err}_t)$, which controls the update direction.
>
> First, RMD only uses the input prompt at the current time step, enabling adaptation before any output is generated. Uncertainty features require completing generation and scoring first, making them inherently reactive. Second, RMD is computationally efficient (0.07ms overhead, Appendix C.7), while typical uncertainty estimation (e.g., self-consistency sampling) requires multiple generations. Third, RMD directly instantiates the $|\Delta_t|$ term in our theoretical analysis (Theorem 4.5), whereas uncertainty features seem mostly heuristic and lack such theoretical grounding.
>
> > Weakness 1: Assumption 4.4 justification
>
> Motivated by this discussion, we have derived a stronger version of Theorem 4.5 that removes Assumption 4.4(b) entirely. We directly minimize the exact quadratic form of $E\_t[\Phi\_{t+1}]$ (Eq. 35). Under only Assumption 4.4(a), the optimal step size becomes:
> $$\gamma\_t^* \propto \rho\Phi\_t + E\_t[(\alpha - \text{err}\_t)\Delta\_t]$$
> The second term is naturally signed: it amplifies the step size when error feedback aligns with the drift, and attenuates it when they are momentarily anti-aligned, preventing over-correction. Since $\Delta\_t$ is latent, PACE estimates its magnitude via $S_{\text{shift}}$ while the directional component is implicitly handled by $(\alpha - \text{err}_t)$ in the update rule. The original Theorem 4.5 is recovered as a special case. The algorithm (Eq. 14) and all experiments are unchanged.
>
> >Weakness 2: Need more adversarial synthetic settings
>
> Following the reviewer's suggestion, we define three regimes using Beta distributions with symmetric difficulty offsets around the calibration distribution Beta(2,2): Easy ($Y \sim \text{Beta}(2,3)$), Medium ($Y \sim \text{Beta}(2,2)$), and Hard ($Y \sim \text{Beta}(3,2)$). Each regime has a distinct prompt embedding center to enable proactive detection. The stream alternates among the three regimes every 500 steps over $T=6000$. The figure (Fig.5) is provided in: https://anonymous.4open.science/r/109_rebuttal-181B/ICML2026_Rebuttal_figs.pdf
>
> | Method | RMSE| Recovery| Coverage | Worst Violation|
> |---|---|---|---|---|
> | SCP | 0.0751 | 85.05 | 0.9000 | 0.2332 |
> | ACI ($\gamma=0.01$) | 0.0450 | 65.78 | **0.9001** | 0.2360 |
> | ACI ($\gamma=0.2$) | 0.0313 | **4.68** | 0.8787 | 0.1941 |
> | DtACI | 0.0376 | 38.13 | 0.8970 | 0.2027 |
> | **PACE** | **0.0300** | 19.93 | 0.8979 | **0.1889** |
>
> PACE achieves the lowest RMSE and recovers faster than the strongest adaptive baseline DtACI, while maintaining valid coverage near the 0.9 target. We also report the Worst Violation, defined as the maximum deviation of the local error rate from the target α, where PACE again achieves the best result. These results confirm that PACE generalizes beyond Gaussian settings to adversarial distributions.
>
> > Key Question1: Sensitivity to c₁ and c₂
>
> The ablation studies on c₁ and c₂ have been provided in Appendix C.4 (Figures 3-4). PACE outperforms all baselines across nearly the entire range. We recommend starting with $c_1 = c_2 = 0.5$ and increase $c_2$ for abrupt shifts or $c_1$ for gradual drift.
>
> > Key Question2: More models
>
> We have added Llama-2-70B as an additional base model on MMLU. Results below:
>
> | Method | Factuality (%) | RMSE | Recovery | Set Size |
> | :--- | :--- | :--- | :--- | :--- |
> | SCP | 90.36 | 0.0081 | 283.79 | 1.58 |
> | ACI ($\gamma$=0.01) | **89.99** | 0.0058 | 85.66 | **1.55** |
> | ACI ($\gamma$=0.1) | 89.32 | 0.0072 | **22.14** | 1.73 |
> | DtACI | 90.05 | 0.0058 | 76.88 | **1.55** |
> | PACE | 89.94 | **0.0037** | 28.01 | 1.64 |
>
> PACE achieves the lowest RMSE (0.0037), reducing tracking error by 36% relative to DtACI and ACI($\gamma$=0.01).
>
> > Missing limitations discussion
>
> 1. PACE currently assumes that atomic claims within a single response are evaluated independently, which is violated in multi-step reasoning where errors propagate across steps.
> 2. PACE is designed for knowledge-intensive text tasks. Extending to multimodal LLMs where factuality spans visual grounding and cross-modal consistency introduces new dimensions beyond the current framework.
> 3. Like all ACI-family methods, PACE provides asymptotic coverage convergence. Non-asymptotic bounds under arbitrary shifts remain an open challenge.

---

> > ### Author Rebuttal · Reviewer_Feh9 · 2026-04-02
> >
> > I appreciate the detailed feedback and will retain my current score.

---

> > > ### Author Response · Authors · 2026-04-05
> > >
> > > Thank you again for your time and the constructive feedback you provided. We deeply appreciate your effort in reviewing our paper and rebuttal.

---

### Official Review · Reviewer_qV1y · 2026-03-13

**Soundness:** 3
**Presentation:** 3
**Significance:** 3
**Originality:** 3
**Overall Recommendation:** 4
**Confidence:** 3

**Summary:**

The paper addresses the challenge of hallucinated outputs in Large Language Models (LLMs) by improving the reliability guarantees provided by conformal prediction in online settings. Existing conformal methods rely on the exchangeability assumption, which is often violated when user queries and interests shift over time. To address this issue, the authors propose PACE, a framework that dynamically updates the target miscoverage parameter using an adaptive step size to maintain valid coverage under distribution shifts. PACE leverages two complementary signals: a proactive shift detection mechanism to estimate the magnitude of distribution changes and a reactive error signal that adjusts updates based on the local coverage gap. Experiments on synthetic and real-world datasets show that PACE consistently outperforms strong adaptive baselines, reducing deviation from the target error rate by up to 60% in QA tasks and achieving 2.5× faster coverage recovery during abrupt shifts while maintaining utility and stability.

**Compliance With Llm Reviewing Policy:**

Affirmed.

**Key Questions For Authors:**

It would be beneficial to include a detailed analysis of the computational complexity of the PACE as well as an evaluation of its performance in practical applications, particularly regarding resource consumption under high-frequency distribution shifts.

**Limitations:**

yes

**Strengths And Weaknesses:**

Strengths:

1. The paper proposes a novel algorithm, PACE, which combines proactive distribution shift detection with reactive error feedback, significantly improving adaptability in online environments with distribution shifts.

2. The method is extensively validated on both synthetic and real-world datasets. Experimental results show that PACE achieves 2.5× faster recovery than baseline methods under abrupt distribution shifts and reduces error rate deviation by 60%, demonstrating strong practical effectiveness.

3. The paper is clearly written and well organized.

Weaknesses:

1. Although the paper claims that PACE is applicable to arbitrary distribution shifts, the complexity of real-world distributions (e.g., nonlinear changes or mixture distributions) may degrade the algorithm’s practical performance, which is not thoroughly discussed in the paper.

2. Although PACE demonstrates strong empirical performance, the paper does not provide a detailed analysis of its computational cost in large-scale real-time applications, particularly the overhead introduced by proactive shift detection and dynamic step-size adjustment.

---

> ### Author Rebuttal · Authors · 2026-03-30
>
> We thank the reviewer for raising the important questions.
> > Weakness 1: Complex distributions may degrade performance
>
> First, we clarify that PACE's coverage guarantee (Theorem 4.2) holds under *arbitrary* distribution shifts with no structural assumptions. To empirically validate performance under complex distributions, we have conducted a new synthetic experiment on a mixture distribution.
>
> We define three Gaussian components with increasing difficulty:
> - Easy ($Y \sim N(-0.3, 1)$, $\alpha^* > 0.1$)
> - Medium ($Y \sim N(0, 1)$, $\alpha^* \approx 0.1$)
> - Hard ($Y \sim N(0.7, 1)$, $\alpha^* < 0.1$),
> each with distinct prompt embeddings. At each time step, a sample is drawn from the mixture $w_t^{\text{easy}} \cdot \text{Easy} + w_t^{\text{med}} \cdot \text{Medium} + w_t^{\text{hard}} \cdot \text{Hard}$, where the weights undergo abrupt regime transitions every 1000 steps (specifically $(0.1, 0.8, 0.1) \to (0.1, 0.1, 0.8) \to (0.8, 0.1, 0.1) \to \cdots$). This creates shifts where $\alpha^*_t$ both increases and decreases, simulating realistic deployment where query difficulty fluctuates. We also include SAOCP (Improved Online Conformal Prediction via Strongly Adaptive Online Learning, ICML 2023), a competitive baseline added per Reviewer VfHU's suggestion, for completeness. Figures for this experiment (Fig. 4) are available at:  https://anonymous.4open.science/r/109_rebuttal-181B/ICML2026_Rebuttal_figs.pdf. We additionally report Worst Violation (maximum deviation of the local error rate from the target $\alpha$).
>
> | Method | Coverage | RMSE | Recovery | Worst Violation |
> | :--- | :--- | :--- | :--- | :--- |
> | SCP | 0.8539 ± 0.0049 | 0.0967 ± 0.0054 | 272.79 | 0.3120 ± 0.1628 |
> | ACI ($\gamma$=0.01) | **0.9002 ± 0.0005** | 0.0266 ± 0.0020 | 25.11 | 0.2009 ± 0.2077 |
> | ACI ($\gamma$=0.2) | 0.8648 ± 0.0018 | 0.0460 ± 0.0021 | 92.49 | 0.1997 ± 0.2006 |
> | DtACI | 0.8972 ± 0.0014 | 0.0302 ± 0.0016 | 26.05 | 0.2079 ± 0.2043 |
> | SAOCP | 0.8879 ± 0.0019 | 0.0222 ± 0.0018 | 2.51 | 0.4389 ± 0.1268 |
> | **PACE (Ours)** | 0.8940 ± 0.0013 | **0.0191 ± 0.0013** | **1.23** | **0.1827 ± 0.2159** |
>
> PACE achieves the lowest RMSE (0.0191), reducing tracking error by 37% relative to the strongest baseline (ACI $\gamma$=0.01, 0.0266), while achieving near-instantaneous recovery (1.23 steps vs. 25+ for reactive methods). This confirms PACE's effectiveness, even under nonlinear mixture dynamics. We will include this experiment in the revised paper.
>
> > Weakness 2: Computational cost analysis missing
>
> We would like to clarify that a detailed latency analysis has been provided in Appendix C.7 (Table 6).
> To summarize, the total latency is dominated by LLM generation and scoring (~16.47s/query). Importantly, this cost is inherent to the base Conformal Factuality pipeline (generation, claim decomposition, and retrieval-based scoring) and is shared by all methods including SCP and not introduced by PACE. The choice of a lighter scoring function (e.g., direct softmax probabilities as in MMLU) would substantially reduce this base cost. PACE's own algorithmic overhead (RMD computation and dynamic update) is merely 0.0724 ms (<0.00001% of total time), computationally on par with DtACI (0.0731 ms). We will add a prominent reference to this appendix in the main text.

---

> > ### Author Rebuttal · Reviewer_qV1y · 2026-04-03
> >
> > I appreciate the authors’ feedback and the effort they made to address my concerns.
> > After carefully considering the rebuttal, I will retain my current score.

---

> > > ### Author Response · Authors · 2026-04-05
> > >
> > > Thank you again for your time and the constructive feedback you provided. We deeply appreciate your effort in reviewing our paper and rebuttal.

---

### Official Review · Reviewer_VfHU · 2026-03-24

**Soundness:** 3
**Presentation:** 4
**Significance:** 3
**Originality:** 3
**Overall Recommendation:** 5
**Confidence:** 3

**Summary:**

This paper proposes an online conformal prediction method applied to LLMs particularly, an extension of Conformal Factuality to the online setting under arbitrary distribution shift at test time. I understood their contribution to be particularly in proposing an algorithm to adaptively adjust the learning rate in now well-known ACI style online conformal prediction updated. They specially consider a LLM-motivated setting where the distribution of prompt and responses can vary arbitrary and extend the conformal factuality algorithm to the online setting and show that their online update performs better than some alternative online-update style method in this setting. They draw theoretical intuitions for their updates and modify the step size based on the what they call (1) proactive shift detection for the magnitude of the distribution shift, (2) a reactive error based on the local coverage gap. They show empirically that their method reacts better to distribution shifts as well as tracks the target coverage level in a closer and more stable fashion during online deployment.

**Compliance With Llm Reviewing Policy:**

Affirmed.

**Final Justification:**

Concerns are addressed. I keep my score.

**Key Questions For Authors:**

1. Please see weaknesses above
2. Typically one of the drawbacks of the learning rate in general ( adaptive or fixed) is its potential effect on the stability or volatility of the set sizes. Could you compare your method against baselines, or just for your own method show how the set sizes are affected in the case of rapid distribution shifts. I would expect since your method uses the covariate signals for the adaptive tuning of the step size, you would be able to see improvements in terms of the volatility of the prediction set sizes.

**Limitations:**

Yes

**Strengths And Weaknesses:**

STRENGTHS:
1. The problem the authors address is well motivated and practical. Arbitrary distributions shifts arise naturally specially in the context of applying CP to LLMs as the stream of user queries can arbitrary vary in many ways.
2.The organization of the paper from the problem formulation, motivation to the theoretical findings followed by the practical estimation of the theoretical components in the methodology section are very well structured.
3. The theoretical justification that the optimal learning rate scales linearly with the current parameter error and the shift magnitude is sound and intuitively useful.
4. The experiments are comprehensive and I appreciate both synthetic and real world datasets used to convey the findings. Furthermore, the Figure 3 and 5 in the appendix are empirically useful findings and highlight the practicality of their method better, as well as answer some of the questions one might have about the tuning of the parameters c1 and c2.


WEAKNESSES:

1. There is a gap between the theoretical findings to the practical estimations of the optimal step size. Particularly there is a loss of generality when introducing the parameters c1, and c2, and well as the sliding window n.
                     1a. With respect to the parameters c1 and c2, the authors earlier in the paper claim that their method significantly eliminates the need for dataset specific tuning due to the online guarantees, and while this is true, the fact that one has to tune the parameters c1 and c2 for a specific dataset, and there are regimes where combinations of these two parameters can perform worse than the baselines as seen in the appendix ( the curve in terms of RMSE is not always below the other baselines in some small regimes and datasets), weakens the authors earlier claim.
                     1b.The effect of the sliding window n is not empirically studied or expanded upon, which also creates confusion and broadens the gap between the practical implementation and methodology and the theoretical findings.
2. For a more comprehensive evaluation, there are more adaptive baselines out there that the authors did not compare against. Could the authors validate the empirical superiority of the adaptive baseline chosen or cite references for where this is validated.

3. Expanding on the point 2, the literature review does not explain the adaptive methods that are explored in the literature, earlier in the paper specifically line 162 is rather misleading as it gives the perception that the only viable baseline is a fixed step size. I understand that in line 144-148 second column the distinction between the authors contribution and existing adaptive methods is for the first time clarified, but I believe a more coherent narrative would strengthen the paper.

4. Again with respect to the claim of line 160 first column, there are conformal prediction methods, ( not just conformal factuality, but rather variants like conformal language modeling (Quach et. al, 2023 ) and Conformal Prediction Beyond the Seen ( 2025) , that explicitly address the unconstrained output space, and ACI can directly be applied to Conformal Prediction Beyond the Seen ( 2025 ) as an online variant which does take into account the unconstrained output space.), thus I found this claim rather confusing on how its directly related to the ACI method. Could you clarify if I am missing something here ? I believe highlighting point 2 ( with the clarification as said above wrt to adaptive methods ) , and 3 only would  not take away from the contribution of the paper and would eliminate confusion.

5. If it’s possible, to extend the claim in line 183 first column ( the convergence rate) either empirically or with further remarks, even in the appendix, it would significantly strengthen this point which I believe is an important contribution.

6. Please mention in line 257 that c1, and c2 are between [0,1] ( Please let me know if this is not true, which would raise further questions).

---

> ### Author Rebuttal · Authors · 2026-03-30
>
> We sincerely thank the reviewer for the thorough evaluation and constructive feedback.
> > Weakness 1a: Gap between theory and practice regarding c₁, c₂
>
> Our claim "eliminates the need for fragile hyperparameter tuning" (L911) referred to ACI's fixed step-size $\gamma$, not $c_1, c_2$. We agree the phrasing was too broad and will revise it for precision.
>
> > Weakness 1b: Effect of sliding window n
>
> We have conducted a new ablation study varying $n \in \\{5, 10, 25, 50, 100\\}$ on synthetic streams.
> | Window Size | Coverage | RMSE | Recovery |
> | --- | --- | --- | --- |
> | $n=5$ | 0.8931 | 0.0467| 38.20 |
> | $n=10$ | 0.8911 | 0.0422 | 55.91 |
> | $n=25$ | 0.8943 | 0.0401| 50.22 |
> | $n=50$ | 0.8966| 0.0433| 62.32 |
> | $n=100$ | 0.8972| 0.0496| 80.27 |
>
> | Window Size | Coverage | RMSE |
> | --- | --- | --- |
> |$n=5$| 0.8963| 0.0279|
> |$n=10$| 0.8929| 0.0252|
> |$n=25$| 0.8947| 0.0238|
> |$n=50$| 0.8974 | 0.0244|
> |$n=100$| 0.8986| 0.0265|
> The results reveal small windows (e.g., $n=5$) overreact to aleatoric noise, while large windows (e.g., $n=100$) introduce adaptation lag. Setting $n=25$ achieves the optimal balance. Crucially, even at sub-optimal configurations, PACE consistently outperforms other baselines as shown in Fig.1 in the: https://anonymous.4open.science/r/109_rebuttal-181B/ICML2026_Rebuttal_figs.pdf. We will include them in the revised Appendix.
>
> > Weakness 2: Incomplete adaptive baselines
>
> We initially chose DtACI because Gibbs & Candès (2024) extensively validated that it consistently outperforms prior adaptive variants (including AgACI and MVP). Thus, it serves as a strong representative of state-of-the-art reactive methods.
> To further strengthen our paper, we have now added SAOCP (Improved Online Conformal Prediction via Strongly Adaptive Online Learning, ICML 2023), a newer and highly competitive baseline. The following are the updated results on synthetic dataset (Fig.2 in: https://anonymous.4open.science/r/109_rebuttal-181B/ICML2026_Rebuttal_figs.pdf). We also report Worst Violation defined as the maximum deviation of the local error rate from the target $\alpha$ across all time.
>
> (a) Abrupt: SAOCP's interval-experts quickly discard pre-shift history, yielding lower recovery time. However, its reactive nature leads to worst-case violations of (.455 vs PACE .250) at shift onset and overall under-coverage (.885 vs .895) due to post-shift volatility.
>
> |Method|Coverage|RMSE|Recovery| Worst Violation |
> | :--- | :--- | :--- | :--- | :--- |
> | DtACI | 0.8862| 0.0461| 120.66 | 0.2754|
> | SAOCP | 0.8857| 0.0293| 21.11 | 0.4556|
> | PACE | 0.8953| 0.0345| 54.18 | 0.2507|
>
> (b) Smooth: Continuous drift exposes SAOCP's reaction lag further; proactive PACE strictly outperforms it across all metrics.
>
> | Method | Coverage | RMSE | Worst Violation |
> | :--- | :--- | :--- | :---|
> | DtACI | 0.8948 | 0.0340| 0.2305|
> | SAOCP | 0.8873| 0.0267| 0.4556|
> | PACE | 0.8982| 0.0233| 0.2232
>
> (c) Mixture: We also outperform SAOCP under a new mixture distribution provided in the response to Reviewer qV1y weakness 1.
>
> > Weakness 3: Misleading narrative in literature review
>
> Thanks for raising this concern. We will revise Section 3.3 to explicitly acknowledge existing adaptive variants immediately after introducing the fixed-step ACI (Eq. 3)
>
> > Weakness 4: Claim about unconstrained output space
>
> ACI can be directly integrated with methods addressing unconstrained output spaces. We will remove the claim as a limitation of ACI and focus our narrative strictly on its fundamental drawback: the reactive nature and suboptimal step-size adaptation. We will also ensure the suggested literature is explicitly discussed in our Related Work section.
>
> > Weakness 5: Convergence rate claim
>
> Thank you for your suggestion. In the original submission, we have validated this empirically in Appendix C.6 (Fig. 6), where the long-term cumulative error rate on WikiData shows PACE converges significantly faster with tighter variance bounds. We will add a direct forward-reference from Remark 4.3 to this appendix.
>
> > Weakness 6: Specify c₁, c₂ range
>
> We’ll mention that c1, c2 are between [0,1].
>
> > Key Question: Prediction set size volatility
>
> We plotted the moving average of the prediction set size on the MMLU stream shown in Fig.3: https://anonymous.4open.science/r/109_rebuttal-181B/ICML2026_Rebuttal_figs.pdf. During abrupt OOD shifts, a conformal predictor must expand its set size to maintain coverage. Conservative methods (DtACI, ACI $\gamma=0.01$) underreact during shifts (peaking only at \~1.4), directly causing the severe error rate increases seen in Figure 2. Aggressive methods (ACI $\gamma=0.1$) overreact, keeping set sizes unnecessarily inflated (>1.2) even during stable periods, degrading utility. Our method sharply expands the set size only during OOD blocks (~1.6) to guarantee safety, then instantly shrinks it to ~1.0 during stable periods. PACE's set size tracks the actual shift magnitude: expanding during OOD blocks and contracting during stable periods.

---

> > ### Author Rebuttal · Reviewer_VfHU · 2026-04-07
> >
> > Concerns are addressed. I keep my score.

---

> > > ### Author Response · Authors · 2026-04-08
> > >
> > > Thank you again for your time and the constructive feedback you provided. We deeply appreciate your effort in reviewing our paper and rebuttal.

---

### Decision · Program_Chairs · 2026-04-30

**Decision:**

Accept (regular)

**Comment:**

This paper proposes an novel online conformal prediction method which dynamically updates the target miscoverage parameter in ACI along with its theoretical justification; the method achieves a better convergence speed compared to baselines.

All reviewers vote for acceptance -- reviewers appreciate this paper and the raised concerns are addressed during the rebuttal.